# Achieving Near-Optimal Individual Regret & Low Communications in Multi-Agent Bandits

**Xuchuang Wang**
Department of Computer Science and Engineering
The Chinese University of Hong Kong
xuchuangw@gmail.com

**Lin Yang**
School of Intelligence Science and Technology
Nanjing University
linyang@nju.edu.cn

**Yu-zhen Janice Chen**
College of Information and Computer Sciences
University of Massachusetts Amherst
yuzhenchen@cs.umass.edu

**Xutong Liu**
Department of Computer Science and Engineering
The Chinese University of Hong Kong
liuxt@cse.cuhk.edu.hk

**Mohammad Hajiesmaili & Don Towsley**
College of Information and Computer Sciences
University of Massachusetts Amherst
{hajiesmaili, towsley}@cs.umass.edu

**John C.S. Lui**
Department of Computer Science and Engineering
The Chinese University of Hong Kong
cslui@cse.cuhk.edu.hk

## Abstract

Cooperative multi-agent multi-armed bandits (`CMA2B`) study how distributed agents cooperatively play the same multi-armed bandit game. Most existing `CMA2B` works focused on maximizing the group performance of all agents—the accumulation of all agents' individual performance (i.e., individual reward). However, in many applications, the performance of the system is more sensitive to the "bad" agent—the agent with the worst individual performance. For example, in a drone swarm, a "bad" agent may crash into other drones and severely degrade the system performance. In that case, the key of the learning algorithm design is to coordinate computational and communicational resources among agents so to optimize the individual learning performance of the "bad" agent. In `CMA2B`, maximizing the group performance is equivalent to minimizing the group regret of all agents, and maximizing the individual performance can be measured by minimizing the maximum (worst) individual regret among agents. Minimizing the maximum individual regret was largely ignored in prior literature, and currently, there is little work on how to minimize this objective with a low communication overhead. In this paper, we propose a near-optimal algorithm on both individual and group regrets, in addition, we also propose a novel communication module in the algorithm, which only needs $O(\log(\log T))$ communication times where $T$ is the number of decision rounds. We also conduct simulations to illustrate the advantage of our algorithm by comparing it to other known baselines.

## 1 Introduction

The stochastic multi-armed bandit problem is a classic sequential decision making problem. Given $K$ arms, there is one agent who repeatedly chooses one arm to pull and observes a stochastic reward from the pulled arm in each time slot. To maximize cumulative reward (or minimize *regret* which is the cumulative reward difference between the optimal decision and agent's choices), the agent needs to pull an arm either with a large empirical mean reward to greedily maximize reward (exploitation), or whose reward estimate is highly uncertain so as to reduce that uncertainty to discover good arms (exploration). To model many real life applications, e.g., cognitive radio with multiple users (Liu & Zhao, 2010; Jouini et al., 2010; Boursier & Perchet, 2019), clinical trials in multiple labs (Wang, 1991), recommendation systems with multiple servers (Agarwal et al., 2008; Li et al., 2010; Landgren et al., 2016), cooperative source search by multiple robots (Li et al., 2014; Jin et al., 2017), etc., one

Table 1: A comparison summary of prior literature and this work (all regret bounds are problem-dependent and we omit the $1/\Delta_2$ factor.)

|  | Individual regret | Group regret | Communication time |
|---|---|---|---|
| DPE2 (Wang et al., 2020a) | $O(K \log T)$ | $O(K \log T)$ | $O(K^2 M^2)$ |
| ComEx (Madhushani & Leonard, 2021) | $O(K \log T)$ | $O(K \log T)$ | $O(KM \log T)$ |
| GosInE (Chawla et al., 2020) | $O((K/M + 2) \log T)$ | $O((K + 2M) \log T)$ | $\Omega(\log T)$ |
| Dec_UCB (Zhu et al., 2021a) | $O((K/M) \log T)$ | $O(K \log T)$ | $O(MT)$ |
| UCB-TCOM (our algorithm) | $O((K/M) \log T)$ | $O(K \log T)$ | $O(KM \log(\log T))$ |

needs to extend the model to allow for more than one agent ($M > 1$) playing the same multi-armed bandit game. These agents cooperate with each other to minimize their regrets. We call this problem the cooperative multi-agent multi-armed bandits (CMA2B) problem and present it formally in §2.

The most common objective of CMA2B is to minimize the aggregate regret among all $M$ agents, dubbed as *group regret* in this paper. This objective has been studied in the majority of prior work (Boursier & Perchet, 2019; Chawla et al., 2020; Huang et al., 2021; Shi et al., 2021b; Wang et al., 2020a;b). In addition to group regret, individual performance among agents is another important metric that is less studied in prior work on CMA2B. The performance of each individual agent is critical in many applications in distributed systems. For example, in many distributed resource allocation scenarios with different agents in charge of the allocation, overall performance depends on the performance of the *bottleneck agent* instead of the aggregate performance of all agents. This can also be seen in a computer network scenario, in which an ISP may apply learning-based algorithms (Ma et al., 2010; Jiang et al., 2018) for some networking problems such as shortest path routing, channel selection, etc. To ensure that the users are served fairly, the underlying algorithms should fairly provide approximately equivalent individual performance for each learning agent. This is equivalent to minimizing the bottleneck agent's individual regret. For another thing, in network optimization literature, the max-min fairness metric—maximize the minimal individual reward—is widely used to measure a system's fairness (Srikant & Ying, 2013, §2.21), such as fair queuing (Demers et al., 1989). Since the regret is the opposite of reward, optimizing the max-min fairness is also equivalent to minimizing the bottleneck agent's regret. Other fairness motivation examples can be found in political philosophy (Rawls, 2004). In this paper, we explicitly take into account the notion of minimizing the *maximum individual regret* and, for brevity, hereinafter, refer to it as the *individual regret*.

Another important metric in CMA2B is the *communication time* of all agents. For some distributed systems, e.g., agents are geographically located, communications among agents can be expensive. Thus it is important to design a cooperative learning algorithm that provides minimal group and individual regrets, while at the same time, incurs a small communication cost. In addition, it will be desirable to have a learning algorithm in which one can *tune parameters* so as to trade off communication times with regret as needed by different applications.

**Contributions.** In §3, we present the UCB-TCOM algorithm that achieves not only a near-optimal group regret of $O((K/\Delta_2) \log T)$ but also a near-optimal problem-dependent individual regret of $O((K/M\Delta_2) \log T)$ with only $O(\log(\log T))$ communication times, where $\Delta_2$ is the smallest reward mean gap between arms and $T$ is the number of rounds. This is the first near-optimal algorithm on individual regret with efficient communications: Previous low communication algorithms, e.g., the leader-follower algorithm (Wang et al., 2020b), cannot achieve the near-optimal individual regret; and previous near-optimal algorithms on individual regret, e.g., GosInE (Chawla et al., 2020), required high communication times (see related works below). UCB-TCOM achieves the near-optimal individual regret performance by evenly dividing the group regret to all agents. To equalize the regrets of all agents, UCB-TCOM directs agents to synchronously pull arms: agents only utilize the common reward observations, i.e., those having been broadcast over all agents, to make decisions. The communication policy TCOM (Tunable COMmunication) of UCB-TCOM is a parametric meta-algorithm that governs the communication of agents and can be executed on top of any underlying bandit learning algorithm. A salient feature of TCOM is that it can be tuned to balance regret and communication times. In particular, two parameters in TCOM can be tuned to determine the aggressiveness and frequency of communications among agents. Our analysis explicitly demonstrates how communication times can be tuned from $0$ to $O(T)$. Finally, we report numerical results in §5.

**Related works.** The most relevant prior literature on `CMA2B` is summarized in Table 1. First of all, notice all algorithms in Table 1 attain a similar $O(K \log T)$ regret upper bound. So, we compare them from the perspective of individual regret and communication cost. Previous algorithms exhibit good individual regret bounds, i.e., `Dec_UCB` (Zhu et al., 2021a) and `GosInE` (Chawla et al., 2020). However, the communication times of `Dec_UCB` are $O(T)$ and that of `GosInE` are at least $\Omega(\log T)$. It is worth noting that the leader-follower algorithm (`DPE2`) (Wang et al., 2020a), where an agent, i.e., the leader, takes responsibility for exploration, and the other agents, i.e., followers, only exploit the arms recommended by the leader incurs constant $O(1)$ communication cost. However, the leader-follower approach inherently fails to achieve good individual regret—the leader agent incurs almost all of the regret $O(K \log T)$ while followers only incur constant regrets. Lastly, we mention the communication policy `ComEx` (Madhushani & Leonard, 2021) designed for fully distributed algorithms achieves a near-optimal group regret of $O(K \log T)$ with a communication cost of $O(\log T)$. However, `ComEx` does not guarantee optimal individual regret. We provide a comprehensive discussion of related work in Appendix A.

## 2 MODEL

**System Model.** Consider a multi-armed bandit (MAB) game with $K \in \mathbb{N}_+$ arms. Each arm $k \in [K]$[1] is associated with a $(1/2)$-sub-gaussian reward distribution with unknown mean $\mu(k)$, e.g., it can be any distribution whose support is $[0, 1]$. Assume there are $M \in \mathbb{N}_+$ distributed agents playing this bandit game in $T \in \mathbb{N}_+$ rounds. In time slot $t \in [T]$, each agent $i \in [M]$ pulls an arm $A_t^{(i)} \in [K]$ and receives a reward $X_t^{(i)}(A_t^{(i)})$ drawn from the reward distribution of arm $A_t^{(i)}$. When more than one agent pulls the same arm, each of them gets an independent reward drawn from the arm's distribution. Without loss of generality, assume the $K$ arms are ordered in descending of their mean rewards, i.e., $\mu(1) > \mu(2) \geqslant \ldots \geqslant \mu(K)$. Denote the reward mean gap as $\Delta_k := \mu(1) - \mu(k)$ and assume $\Delta_2 > 0$, i.e., arm 1 is the unique optimal arm.

**Group regret.** This paper uses group regret—the cumulative difference between the optimal policy rewards and an algorithm's rewards—as the algorithm's performance metric. Each agent's optimal policy is to pull arm 1 in all $T$ rounds. So, under any cooperative algorithm $\mathcal{A}$, the expected group regret of all $M$ agents is defined as follows,

$$\mathbb{E}[\mathrm{R_T}(\mathcal{A})] := MT\mu(1) - \mathbb{E}\left[\sum_{i \in [M]} \sum_{t \in [T]} X_t^{(i)}(A_t^{(i)})\right], \tag{1}$$

where the expectation is taken over the randomness of stochastic rewards and algorithm's (agents') decisions.

**Maximum individual regret.** While the group regret in (1) characterizes overall system performance, individual performance of each agent is important as well, and, among all individual regrets of agents, the maximum one is usually more important. For example, in a drone swarm, the failure/misbehavior of a single drone, e.g., it crashes into other drones, can dramatically degrade the whole system's overall performance; in network measurement, the slowest inference engine determines how fast the network parameters, e.g., traffic flows and channel bandwidths, are learned. In the above systems, one can define the *individual regret* objective as follows,

$$\mathbb{E}[\mathrm{R_T^{ind}}(\mathcal{A})] := T\mu(1) - \mathbb{E}\left[\min_{i \in [M]} \sum_{t \in [T]} X_t^{(i)}(A_t^{(i)})\right]. \tag{2}$$

**Communication times.** Agents in the system cooperate with each other by communicating their reward observations or reward averages. Any message (if communicated) is broadcast to all agents and received within a single time slot. A broadcast message includes an arm index, the average arm reward for all observations since the previous broadcast, and the number of observations. The total number of messages communicated among these $M$ agents quantifies the communication times of an algorithm. We count each broadcast as one message[2]. Hence, the communication times under

---

[1]In this paper, for any integer $L$ we denote the set $\{1, 2, \ldots, L\}$ as $[L]$.

[2]In this work, we choose the name "communication times" to emphasize that we focus on reducing the number of communicated messages/communication rounds. There are some works considering the bits in communication, e.g., Wang et al. (2020b); Hanna et al. (2021); Shi et al. (2021b). Studying the necessary bits for `TCOM` is an interesting further work, e.g., one can consider apply the adaptive differential communication (Shi et al., 2021b) to our observation sharing.

algorithm $\mathcal{A}$ are defined as follows,

$$\mathbb{E}\left[C_T(\mathcal{A})\right] \coloneqq \mathbb{E}\left[\sum_{i \in [M]} \sum_{t \in [T]} \mathbb{1}\{\text{agent } i \text{ broadcasts one message in time slot } t\}\right]$$

**Model extensions.** For ease of presentation of the core ideas, we focus on a simple model formulation where agents reside on a complete graph (i.e., one clique) and incur no communication delays. In Appendix G, we extend the basic model and communication policy TCOM to account for arbitrary communication topology and deterministic communication delays. These demonstrate the generality of our core algorithmic ideas.

## 3   ALGORITHM DESIGN

In this section, we devise a cooperative learning algorithm, UCB-TCOM, which attains the near-optimal results of not only group regret but maximum individual regret as well with only $O(\log(\log T))$ communication times. The core of UCB-TCOM is a communication strategy TCOM that can run on top of diverse bandit learning algorithms in multi-agent systems. Specifically, TCOM determines how an agent broadcasts its reward observations to other agents, while the underlying bandit learning algorithm, e.g., UCB, Thompson sampling, elimination, determines how each agent pulls arms.[3] In §3.1, we present TCOM as the communication module of our algorithm design. Then, in §3.2, we show how to integrate TCOM with UCB as the underlying bandit algorithm to obtain UCB-TCOM.

Recall that in CMA2B, the near-optimal group regret $\Theta((K/\Delta_2)\log T)$ (Wang et al., 2020a, §1.2) is the summation of individual regrets of all agents. Therefore, minimizing the maximum individual regret is equivalent to equalizing individual regrets of all agents. Uniformly dividing the group regret to each individual agent can be achieved by a symmetric learning structure where agents pull same arms in each time slot. However, in current state-of-the-art algorithms, e.g., the leader-follower algorithm DPE2 (Wang et al., 2020a), agents usually take different roles—some explore and other exploit—and thus those algorithms are suboptimal on the individual regret objective. For another thing, some straightforward symmetric learning algorithms, e.g., cooperative UCB (Yang et al., 2022, §III.B) where agents communicate and take the same action in every time slots, requires high communication times; most are $O(T)$. Instead UCB-TCOM proposes a novel communication policy TCOM to maintain a symmetric learning structure with low communication times. Especially, to maintain the symmetric learning structure, agents only utilize *global* reward observations. Here *"global"* observations refer to those observations that have already been broadcast to all agents, excluding those recent reward observations that have not been shared (i.e., only local to one agent).

### 3.1   TCOM: A TUNABLE COMMUNICATION POLICY

A key idea of TCOM is based on the observation that the benefit of cooperation in a multi-agent setting comes primarily from sharing information regarding suboptimal arms instead of the optimal arm. Specifically, an agent can avoid pulling a suboptimal arm when she receives external observations from others, while sharing observations from the optimal arm may not necessarily reduce individual agent regret, e.g., sharing information about an optimal arm while other agents still need to exploit the optimal arm will not decrease regret but merely increase communication times. So, instead of paying the cost for communicating the optimal arm observations, an agent can let other agents explore the optimal arm by themselves. Hence, agents can reduce overall communication overhead by refraining from communicating the optimal arm's rewards while still guaranteeing near-optimal regret. To implement the above idea, each agent constructs a communication arm set from which TCOM dynamically excludes the arms that are likely to be optimal. Then, each agent only shares observations about potentially suboptimal arms in the communication arm set. We explain how to construct the communication arm set in §3.1.1.

The communication arm set technique in the previous paragraph reduces communication times to $O(\log T)$. To further reduce communication times, another key idea of TCOM is to aggregate an arm's multiple observations and send their average at one time. Naïvely applying a phase-based

---

[3]Besides the UCB-TCOM algorithm presented in this section, we also preset the Thompson sampling and elimination based algorithm in Appendix F

communication protocol, e.g., the one in (Wang et al., 2020b), to aggregate arm observations does not automatically further decrease communication times to $O(\log(\log T))$. Because the phase-based protocol was used to achieve $O(\log T)$ communication times which is already reached via the communication arm set technique. Instead, one needs to break the pre-assigned rigid phased-based communications, i.e., observations of all arms are communicated together at the end of each phase, and allow each arm—depending on how many times it lies inside the communication arm set—to have its own specific and dynamic phases for communications. This idea is implemented by a carefully designed observation-buffering technique. The novel technique can boost the above communication arm set technique and further reduce communication times to $O(\log(\log T))$. We formalize this idea in §3.1.2.

### 3.1.1 CONSTRUCTION OF THE COMMUNICATION ARM SET

In each time slot, each agent determines its communication arm set by comparing its estimated confidence intervals of the reward means. Let $\hat{n}_t(k)$ denote the number of times of the *global* reward observations of arm $k$ up to time slot $t$. That is, the number of observations that has been broadcast across the whole CMA2B. Let $\hat{\mu}_t(k)$ denote the mean reward estimate of these $\hat{n}_t(k)$ observations. Then, given the empirical mean $\hat{\mu}_t(k)$ and the number of observations $\hat{n}_t(k)$, we construct a *tunable* confidence interval for the true reward mean $\mu(k)$ by Hoeffding's inequality (Hoeffding, 1994). The confidence interval centered at $\hat{\mu}_t(k)$ has a width $\mathtt{CI}_t(k, \alpha)$ expressed as

$$\mathtt{CI}_t(k, \alpha) \coloneqq \alpha \sqrt{\log t / \hat{n}_t(k)},$$

where $\alpha \in (-\infty, \infty)$ is a tunable parameter. The reward mean $\mu(k)$ lies inside the confidence interval $(\hat{\mu}_t(k) - \mathtt{CI}_t(k, \alpha), \hat{\mu}_t(k) + \mathtt{CI}_t(k, \alpha))$ with probability of at least $1 - 2t^{-2\alpha^2}$ (see Lemma 1 in Appendix B.2). We define the tunable upper and lower confidence bounds of $\hat{\mu}_t(k)$ as

$$\mathtt{tUCB}_t(k, \alpha) \coloneqq \hat{\mu}_t(k) + \mathtt{CI}_t(k, \alpha), \quad \mathtt{tLCB}_t(k, \alpha) \coloneqq \hat{\mu}_t(k) - \mathtt{CI}_t(k, \alpha).$$

For any arm $k$, if there exists another arm $k'$ whose upper confidence bound $\mathtt{tUCB}_t(k', \alpha)$ is greater than arm $k$'s lower confidence bound $\mathtt{tLCB}_t(k, \alpha)$, then arm $k$'s mean reward estimate is not significantly greater than those of others. In this case, arm $k$ is potentially identified as suboptimal, and its new observation (if any) should be broadcast to others. Formally, given tuning parameter $\alpha$, communication arm set $\mathcal{C}_t(\alpha)$ of agent $i$ at time $t$ contains all arms identified as suboptimal, i.e.,

$$\mathcal{C}_t(\alpha) \coloneqq \{k \in [K] : \exists k' \in [K] \setminus \{k\} \text{ such that } \mathtt{tUCB}_t(k', \alpha) > \mathtt{tLCB}_t(k, \alpha)\}. \tag{3}$$

**Remark 1** (Comparison to the candidate arm set of elimination algorithms). *At the first glance, the communication arm set in (3) is similar to the active arm elimination (AAE) policy's candidate arm set (Auer et al., 2002; Even-Dar et al., 2006; Yang et al., 2022). However, the two sets have intrinsically different usages:* TCOM *constructs the communication arm set to determine how to broadcast observations, while AAE uses its candidate set to determine how to pull arms.*

**Remark 2** (Comparsion to ComEx (Madhushani & Leonard, 2020; 2021)). *The intuition we utilize in this subsection—communicating the observations of suboptimal arms is more useful than that of the optimal arm—has also been observed in Madhushani & Leonard (2020; 2021). Compared to* ComEx, *the algorithmic novelties of* TCOM *are (1) applying confidence intervals to more accurately identify the suboptimal arms; (2) combining the observation-buffering broadcast with the communication arm set technique; (3) including tunable parameters with more flexibility. The detailed algorithm design comparisons are deferred to Appendix A.1. Furthermore, as we mentioned in Table 1,* ComEx *fails to achieve the near-optimal individual regret and its communication times $O(\log T)$ are not as good as our $O(\log(\log T))$ result.*

### 3.1.2 OBSERVATION-BUFFERING BROADCAST

While the straightforward *immediate broadcast* observation sharing method can produce a near-optimal group regret, its $O(\log T)$ communication times are not efficient. Prior literature (Desautels et al., 2014; Gao et al., 2019) has shown that the near-optimal regret in stochastic bandit games is preserved in the presence of a constant delay before receiving reward observations (Desautels et al., 2014), and the regret upper bound does not deteriorate too much when delays increase according to a geometric sequence (Gao et al., 2019). These results suggest that it is not necessary to immediately communicate the latest observations. Instead, to reduce communication times, the

---

**Algorithm 1** The `UCB-TCOM` Algorithm (for each agent)

---

1: **Input:** the communication arm set parameter $\alpha$ and buffering ratio $\beta$
2: **Initialization:** $\hat{n}_t(k) = 0, N_t(k) = 0, \hat{\mu}_t(k) = 0, \tau_t(k) = 0$
3: **for** each decision round $t$ **do**  ▷ Both for-loops (Lines 3 and 15) run in parallel.
4:    Pull arm $A_t$ with the highest *global* UCB
5:    Observe arm $A_t$'s reward $X_t(A_t)$
6:    **if** $A_t \in \mathcal{C}_t(\alpha)$ **then**
7:       Increase $N_t(A_t)$ by 1
8:       Update this phase's empirical mean $\tilde{\mu}_t(A_t)$
9:    **end if**
10:    **if** $N_t(A_t) \geqslant \lceil \beta N_{\tau_t(A_t)}(A_t) \rceil$ **then**
11:       Broadcast the message $(\tilde{\mu}_t(A_t), N_t(A_t), A_t)$
12:       $\tau_t(A_t) \leftarrow t$
13:    **end if**
14: **end for**
15: **for** each newly received message $(\tilde{\mu}_t(k), N_t(k), k)$ from the past decision round **do**
16:    Update the empirical mean $\hat{\mu}_t(k) \leftarrow \frac{\hat{\mu}_t(k)\hat{n}_t(k) + \tilde{\mu}_t(k)\lfloor N_t(k)(1-1/\beta) \rfloor}{\hat{n}_t(k) + \lfloor N_t(k)(1-1/\beta) \rfloor}$
17:    Increase $\hat{n}_t(k)$ by $\lfloor N_t(k)(1 - 1/\beta) \rfloor$
18:    Update the communication arm set $\mathcal{C}_t(\alpha)$ via (3) based on tunable confidence bounds
19: **end for**

---

policy can aggregate multiple reward observations from an arm and communicate their sample mean in one message. Combining this mechanism with the designed communication arm set in (3) leads to the communication policy `TCOM`.

Next, we present how the buffering mechanism controls observation aggregation. Denote by $N_t(k)$ the number of times that an agent pulls arm $k$ when the arm lies inside the communication arm set, i.e., $k \in \mathcal{C}_t(\alpha)$, up to time $t$. Therefore, $N_t(k)$ is the number of messages agent $i$ would broadcast, each containing a single arm $k$ observation, if one employs immediate broadcast. Instead of immediate broadcast, we buffer observations and then broadcast their average when the number of observations (since previous broadcast) increases by a ratio. For example, if the ratio is 2, the agent sends arm $k$'s reward average of the arm's new observations since last broadcast whenever the counter $N_t(k)$ equals $2, 4, 8, 16, \ldots$, etc, or power of 2. More generally, let $\beta (> 1)$ denote a ratio parameter that controls how the buffering size of reward observations is increased. When counter $N_t(k)$ increases to $\lceil \beta N_\tau(k) \rceil$ since the last time agent $i$ broadcast arm $k$'s average reward, i.e., $N_t(k) \geqslant \lceil \beta N_\tau(k) \rceil$, then agent $i$ broadcasts a message including the arm index $k$, the counter $N_t(k)$, and the sample mean $\tilde{\mu}_t(k)$, of this arm's latest $\lfloor N_t(k)(1 - 1/\beta) \rfloor$ observations to other agents. Due to the space limit, the detailed discussion of `TCOM`'s tunability on parameters $\alpha$ and $\beta$ is deferred to Appendix D.1.

### 3.2 UCB-TCOM: APPLICATION OF TCOM ON THE UCB ALGORITHM

In this section, we demonstrate how to leverage the communication policy `TCOM` to develop a communication-efficient extension of cooperative `UCB`. We call the proposed algorithm `UCB-TCOM`.

In slot $t$, agent $i$ chooses the arm with the largest *global* UCB index, i.e., $A_t = \arg\max_k \text{UCB}_t(k)$, where $\text{UCB}_t(k) := \hat{\mu}_t(k) + \sqrt{2\log t / \hat{n}_t(k)}$, and observes the arm's reward $X_t^{(i)}(A_t)$. Note the UCB indexes and action $A_t$ are the same for all agents; thus we omit their superscript $(i)$. After pulling arm $A_t$, the agent checks whether the arm is in its communication arm set $\mathcal{C}_t$ (Line 6); if yes, counter $N_t(A_t)$ is increased by 1, and the buffered reward sample mean is updated (Lines 7-8). When the counter $N_t(A_t)$ is increased by a factor $\beta$ since the last time slot that agent $i$ communicated arm $A_t$'s sample mean (Line 10), the agent sends out arm $A_t$'s sample mean to other agents (Line 11). Meanwhile, if the agent receives messages from other agents, it updates its estimates accordingly (Lines 16- 18). The details are presented in Algorithm 1.

## 4 THEORETICAL RESULTS

In this section, we analyze the regret and communication times of `UCB-TCOM` (Algorithm 1).

### 4.1 RESULT OVERVIEW

We present near-optimal group and individual regret upper bounds of `UCB-TCOM` in Theorem 1.

**Theorem 1** (Regret upper bounds of `UCB-TCOM` for $\alpha > 1$). *When the communication arm set parameter $\alpha > 1$[4] and buffering-ratio $\beta > 1$, `UCB-TCOM` (Algorithm 1) attains a near-optimal group regret upper bound in terms of number of decision rounds $T$, arms $K$, and agents $M$, or formally,*

$$\mathbb{E}[\mathrm{R}_\mathrm{T}(\mathcal{A})] \leqslant \sum_{k>1} \frac{8\beta \log T}{\Delta_k} + MK \frac{2\alpha^2 - 1}{\alpha^2 - 1}, \tag{4}$$

*and `UCB-TCOM` (Algorithm 1) also attains a near-optimal individual regret upper bound, or formally,*

$$\mathbb{E}[\mathrm{R}_\mathrm{T}^{\mathrm{ind}}(\mathcal{A})] \leqslant \sum_{k>1} \frac{8\beta \log T}{M\Delta_k} + K \frac{2\alpha^2 - 1}{\alpha^2 - 1}. \tag{5}$$

Theorem 1's group regret proof is based on the observation that most suboptimal arm observations are broadcast when $\alpha > 1$ (see Theorem 2(iii)), and the fact that under the observation buffering mechanism the number of observations of suboptimal arms increases by at most a ratio of $\beta$ between broadcasts. Theorem 1's individual regret proof is based on the fact that all $M$ agents are symmetric—they always pull the same arms. Proofs of Theorem 1 is given in Appendix B.3.

In Theorem 2, we present the communication times of `UCB-TCOM`. It shows how parameter $\alpha$ influences the communication of optimal and suboptimal arms observations.

**Theorem 2.** *The communication times of `UCB-TCOM` has the following properties:*

(i) *When $\alpha \leqslant -\sqrt{2}$, no communication occurs among agents.*

(ii) *When $-\sqrt{2} < \alpha < \sqrt{2}$ and $\beta > 1$, the number of broadcasts of observations of the optimal arm by one agent is $O(\log(\log T))$. More rigorously, it is less than*

$$\log_\beta \left( \left( \frac{\sqrt{2} + \alpha}{\sqrt{2} - \alpha} \right)^2 \left( \frac{8 \log T}{\Delta_2^2} + MK \frac{2\alpha^2 - 1}{\alpha^2 - 1} \right) \right). \tag{6}$$

(iii) *When $\alpha > 1$, almost all observations of suboptimal arms—except for a finite number independent of $T$—are broadcast.*

(iv) *When $\alpha \geqslant \frac{2\sqrt{2}\mu(1)}{\Delta_2}$, almost all observations of the optimal arm—except for a finite number that is independent of $T$—are broadcast.*

The proof of Theorem 2(i) follows from the fact that when $\alpha < -\sqrt{2}$, the pulled arm's `tLCB` index is greater than those `tUCB` indexes of all other arms, which violates the communication arm set's condition in (3). Hence, the pulled arm never lies inside the communication arm set, and thus no communication occurs. The proof of Theorem 2(ii) is based on the geometric growth of the time interval between broadcasts and a key observation: the number of broadcasts of the optimal arm's observations is upper bounded by some suboptimal arm $k$'s total number of pulls, that is, $\hat{n}_t^{(i)}(1) < ((\sqrt{2} + \alpha)/(\sqrt{2} - \alpha))^2 \hat{n}_t^{(i)}(k)$. Theorems 2(iii) and 2(iv) are derived by excluding some well-designed small probability events and from the fact that the arm mean rewards lie inside their tunable confidence intervals with high probability. The proof of Theorem 2 is given in Appendix B.4.

**Remark 3.** *Theorem 2's proof is not a simple extension of the analysis of previous algorithms. First, `TCOM` relies on a group of tunable confidence intervals to identify suboptimal arms and communicate new observations while `ComEx` are only based on arm empirical reward mean; a special case of tunable confidence intervals. Second, the observation-buffering broadcast incurs additional delays of the global observations sharing which is a unique challenge in `TCOM`. We defer the detailed explanation of both challenges to Appendix B.1.*

---

[4]Theorem 1's condition $\alpha > 1$ is a remnant from bounding a small probability event (see Lemma 2 in Appendix B.2). This condition can be relaxed to $\alpha > \frac{1}{\sqrt{2}}$ via the peeling technique (see Appendix C) where (4)'s second term will also change accordingly. This relaxation is also applicable to Theorem 2(iii).

## 4.2 DISCUSSIONS

**(a) Regret optimality of** `UCB-TCOM`**.** The following asymptotic lower bound has been established in the classic bandits literature (e.g., Lai et al. (1985)):

$$\liminf_{T\to\infty} \frac{\mathbb{E}[R_T(\mathcal{A})]}{\log T} \geqslant \sum_{k>1} \frac{\Delta_k}{\mathrm{KL}(\nu_k, \nu_1)}, \tag{7}$$

where $\nu_k$ is the reward distribution of arm $k$ and KL is the Kullback-Leibler divergence between two probability distributions. When the reward distributions are Bernoulli or Gaussian, the regret lower bound can be written as $\Omega(\sum_{k>1}(\log T/\Delta_k))$ (Lattimore & Szepesvári, 2020, §16). Since the `CMA2B` model has the same objective as the centralized (single agent) MAB model, it inherits this group regret lower bound (Wang et al., 2020a, §1.2) under any possible communication policies. Given this lower bound, Theorem 1's group regret upper bound in (7) is near-optimal, i.e., it is tight up to a constant factor.

Recall that the individual regret in (2) corresponds to the maximal individual regret among $M$ agents. Since the maximal individual regret is minimized when all agents uniformly pay the regret time, from the above group regret lower bound, one can obtain the following individual regret lower bound,

$$\liminf_{T\to\infty} \frac{\mathbb{E}[R_T^{\mathrm{ind}}(\mathcal{A})]}{\log T} \geqslant \frac{1}{M}\sum_{k>1} \frac{\Delta_k}{\mathrm{KL}(\nu_k, \nu_1)}.$$

Compared to this lower bound, Theorem 1's individual regret upper bound in (5) is also near-optimal.

**(b) Impact of cooperation and number of agents $M$ on group regret.** If there is no cooperation among the $M$ agents—each agent individually plays the game, then the group regret is lower bounded as follows:

$$\liminf_{T\to\infty} \frac{\mathbb{E}[R_T(\mathcal{A})]}{\log T} \geqslant M\sum_{k>1} \frac{\Delta_k}{\mathrm{KL}(\nu_k, \nu_1)},$$

which is $M$ times worse than the group regret achieved by `UCB-TCOM` in Theorem 1. This highlights the benefits of cooperation. Furthermore, when the number of rounds $T$ is large, the second term containing an $M$ factor in the group regret (4) is negligible. That is, the group regret of `UCB-TCOM` does not increase rapidly when the number of agents $M$ increases, while it would without cooperation.

**(c) Individual regret comparison to the leader-follower algorithm.** Equation (5) shows that the individual regret of `UCB-TCOM` deceases as the number of agents $M$ increases. This is in contrast to `DPE2` (Wang et al., 2020a) based on the leader-follower paradigm, which attributes all exploration costs to a single leader, and thus `DPE2`'s individual regret is $M$ times worse than that of `UCB-TCOM`. In many large-scale distributed systems, e.g., edge computing, database, etc., computational resources are often distributed geographically. The largest individual computational time that these distributed nodes (agents) need to pay can be mapped to the individual regret. From this perspective, `UCB-TCOM` with the near-optimal individual regret is preferable to `DPE2` especially when $M$ is large.

**(d) Communication Times of** `UCB-TCOM`**.** Theorem 2(ii) shows that when $-\sqrt{2} < \alpha < \sqrt{2}$, the number of broadcasts of the optimal arm's observations is $O(\log(\log T))$. Additionally, the observation buffering mechanism also aggregates the suboptimal arms $O(\log T)$ observations into $O(\log(\log T))$ broadcasts. So, in this case, `UCB-TCOM` only incurs $O(\log(\log T))$ communication time. Combined the above with Theorem 1 shows that when $1 < \alpha < \sqrt{2}$ and $\beta > 1$, `UCB-TCOM` enjoys the near-optimal group and individual regret upper bounds with only $O(\log(\log T))$ communication times.

We also discuss how parameters $\alpha$ and $\beta$ of `TCOM` influence cooperation among the agents and how they impact communication times and regret in Appendix D.

## 5 NUMERICAL SIMULATIONS

**Experiment setup.** Unless otherwise stated, the experiments consist of $M = 25$ agents and $K = 20$ arms, communication set parameter $\alpha = 1.2$, buffering ratio $\beta = 2$, and $T = 30,000$. Each arm is

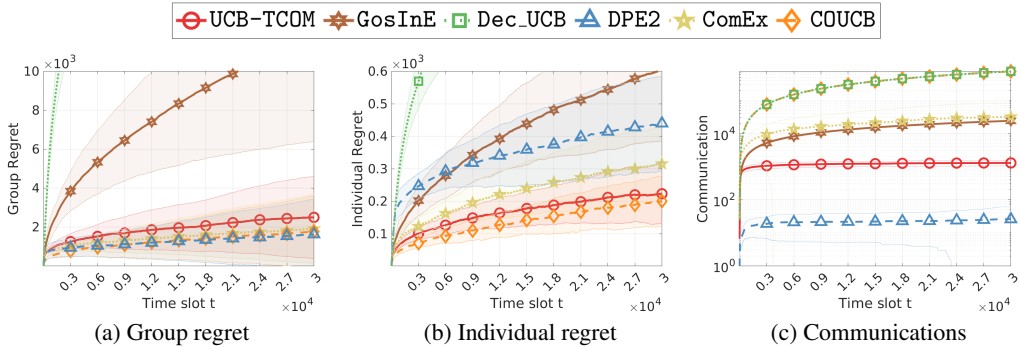

Figure 1: UCB-TCOM vs. Dec_UCB, GosInE, DPE2, ComEx and COUCB

associated with a Bernoulli reward random variable whose mean is uniformly randomly taken from *Ad-Clicks* (Avito, 2015). All results are averaged over 50 trials and their standard deviations are plotted as shaded regions.

**Comparison to state-of-the-art baselines.** We consider five baselines: Dec_UCB (Zhu et al., 2021a), GosInE (Chawla et al., 2020) DPE2 (Wang et al., 2020a), ComEx (Madhushani & Leonard, 2021), and COUCB. The first four are discussed in Table 1. To compare all algorithms fairly, we implement them using the same undirected complete communication graph where agents can broadcast its observations to all other agents. The communication times of GosInE is set to as small as the algorithm allows. COUCB is a naïve adaptation of UCB in CMA2B, where each agent runs an instance of UCB and always immediately broadcasts its new reward observations to other agents.

Comparison results are reported in Figure 1. Both group and individual regrets are reported in Figures 1a and 1b as a function of time. UCB-TCOM performs only slightly worse than the best algorithm (DPE2 in group regrets, COUCB in individual regrets). Figure 1c compares the communication times of these algorithms in the log scale. It corroborates the fact that UCB-TCOM-1's $O(\log(\log T))$ communication times lie between ComEx and GosInE's $O(\log T)$ and DPE2's $O(1)$ performance.

We also provide additional simulations in Appendices D, E and F.

## 6 CONCLUSION

In this paper, we proposed a communication policy TCOM which, combined with diverse single-agent bandit learning algorithms, can be used to devise fully distributed cooperative multi-agent bandit learning algorithms. We specifically study the UCB-TCOM algorithm where agents pull arms via UCB and communicates via TCOM. We show that UCB-TCOM can attain both near-optimal group and individual regrets while only incurring $O(\log(\log T))$ communication costs. Extensive simulations are also reported to support our results.

We note that TCOM can also be tuned to balance the regret and communications. One interesting future work is to systemically study the Pareto frontier of group/individual regrets vs. communication times from the perspective of the upper bounds of TCOM and the intrinsic lower bounds (difficulties) of CMA2B as well.

## 7 ACKNOWLEDGEMENTS

The work of Mohammad Hajiesmaili is supported by NSF CAREER-2045641, CPS-2136199, CNS-2106299, and CNS-2102963. The work of Don Towsley is funded in part by US Army contract W911NF-17-2-0196. The work of John C.S. Lui is supported in part by the RGC's GRF 14215722. Lin Yang is the corresponding author.

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

## A    RELATED WORKS

**CMA2B with a central server or leader.**  Many distributed bandits models (Wang et al., 2020b;a; Shi & Shen, 2021; Shi et al., 2021a; Mehrabian et al., 2020; Shi et al., 2021b) assume the existence of a central server or elect a leader among agents. We choose two mostly related works to discuss. Wang et al. (2020b) consider the CMA2B model with a central server. The server collects information from each agent and then sends its aggregated information back to each agent. Specifically, Wang et al. (2020b) propose a phased-based elimination algorithm that achieves the near-optimal problem-independent regret bound $O(\sqrt{KMT\log(KM)})$ with finite number of communication times (independent of the number of rounds $T$). However, when it comes to near-optimal problem-dependent regret upper bound which is the focus of this paper, their algorithm suffers logarithmic communication times while our algorithm only incurs $O(\log(\log T))$ communications. Wang et al. (2020a) propose the DPE2 algorithm for CMA2B based on a leader-follower paradigm: The leader takes the responsibility of exploring while other agents (followers) only exploit arms recommended by the leader. Although it achieves the optimal group regret with finite number of communications, the DPE2 algorithm is inherently unfair — one single agent, the leader, incurs almost all regret $O(K \log T)$ while other followers only incur finite regrets $O(1)$. We also note that algorithms of previous works heavily rely on a central server (or an elected leader) and thus are vulnerable to attacks on the central server (or a leader), while fully distributed algorithms that this paper studies can tolerate such attacks.

**CMA2B without a central server.**  The fully decentralized CMA2B model has been studied in Buccapatnam et al. (2015); Landgren et al. (2016); Kolla et al. (2018); Martínez-Rubio et al. (2019), where they mainly focus on minimizing regrets and seldom quantify their communication times. Some of these algorithms assume that agents can gossip with each other, e.g., via a graph represented by the gossiping matrix (Martínez-Rubio et al., 2019; Chawla et al., 2020), or via peer-to-peer protocols (Szorenyi et al., 2013). In their gossip setting, agents can only communicate with their neighbors. This is fairly different from our all-agent broadcast communication (in Appendix G.2, we relax the all-agent broadcast in TCOM to communication with neighbors in the graph). Recently, Madhushani & Leonard (2021) proposed a communication protocol, ComEx, which aims to limit the communications of optimal arm's reward observations. Our tunable communication policy TCOM subsumes ComEx as a special case (see Remark 1). Also, ComEx incurs $O(\log T)$ communication times while our algorithm only needs $O(\log(\log T))$ to attain the near-optimal regret upper bound.

Beyond the above related work, there are other works on distributed bandits such as federated bandits (Shi & Shen, 2021; Shi et al., 2021a; Zhu et al., 2021b; Huang et al., 2021), distributed bandits with collision setting (Boursier & Perchet, 2019; Mehrabian et al., 2020; Wang et al., 2020b; Shi et al., 2021b), and cooperative pure exploration (Hillel et al., 2013; Tao et al., 2019; Karpov et al., 2020). The federated bandits (Shi & Shen, 2021; Shi et al., 2021a; Zhu et al., 2021b; Huang et al., 2021) consider heterogeneous local reward distributions, which is different from our homogeneous reward environment. In distributed bandits with collision setting model (Boursier & Perchet, 2019; Mehrabian et al., 2020; Wang et al., 2020b; Shi et al., 2021b), when more than one agent chooses the same arm, each of these agents gets a zero reward, while in our model, each of these agents obtains an independent reward. Cooperative pure exploration (best arm identification) (Hillel et al., 2013; Tao et al., 2019; Féraud et al., 2019; Karpov et al., 2020) study how multi-agent cooperation can reduce the sample complexity of identifying the best arm which is a different objective from our work on regret minimization. Some of their high-level ideas in communication are similar to TCOM, e.g., Hillel et al. (2013) utilize the doubling phase for communication and Féraud et al. (2019)'s algorithm communicates bad arms. However, as we discussed in Appendix B.1, our algorithm is novel in specific design detail, and thus introduces unique challenges that cannot be addressed by the techniques of these known works.

**Individual regret in** CMA2B    With the majority of previous CMA2B works studying the group regret objective, the individual regrets were also studied sparingly, e.g., in stochastic CMA2B (Zhu et al., 2021a; Zhu & Liu, 2021; Zhu et al., 2020), in adversarial CMA2B (Bar-On & Mansour, 2019), and in federated bandits (Zhu et al., 2021b). Among them, Zhu et al. (2021a) was the most related to ours, where they devised algorithms that achieved the near-optimal individual regret with $O(T)$ communication times, while our algorithms, with near-optimal individual regrets as well, only needs $O(\log(\log T))$ communication times.

### A.1 Detail Comparison to ComEx

TCOM shares some high-level ideas from known algorithms. However, when it comes to the algorithm design details, TCOM needs to resolve unique challenges to improve the state-of-the-art result. There are three algorithmic ideas in the design of TCOM which enables improving the state-of-the-art result in the literature. In the following, we outline each idea separately and clarify its novelty as compared to the most relevant prior work.

1. **On how to identify suboptimal arms:** The most related work with similar high-level algorithmic idea is ComEx [Madhushani et. al., 2021] where arms with suboptimal empirical reward means are identified as suboptimal in communication. In TCOM, however, we utilize a group of tunable confidence intervals, which as compared to the empirical means in ComEx, provides a clearer separation between optimal and suboptimal arms. Further, we emphasize that although using confidence intervals in bandits is a classic technique for minimizing regret, TCOM instead uses confidence intervals to decide whether an arm's observation should be shared or not. It is worth noting that the dynamic construction of the communication arm set emphasizes the need for different proof techniques than those in ComEx, which is elaborated in Appendix B.1.

2. **On how to buffer observations:** Observation buffering (a.k.a. phase-based communication) has been used in previous works, e.g., [Shi et. al., 2021], where applying this techniques can achieve a $O(\log T)$ communication times. However, TCOM has its own technique for construction of communication arm sets (relevant to the first algorithmic idea above), which already enjoys a $O(\log T)$ communication times improvement, without using batched communication. Therefore, simply applying the batched communication technique does not automatically improve the communication times of TCOM. Instead, we design an arm-specific observation-buffering mechanism which allows each arm—depending on whether it lies in the communication arm set—to have its own specific and dynamic phases for communication. This carefully designed observation-buffering technique can boost TCOM's first communication arm set technique and further reduce the communication times to $O(\log(\log T))$.

3. **On algorithm design with a tunable parameter:** Last, unique to our algorithm design, we introduced two tunable parameters $\alpha$ and $\beta$ that determine how to add arms into the dynamic communication arm set based on their confidence intervals. This addition provides greater flexibility to TCOM, which is explained in details in Appendix D.

## B  Proofs in Analysis (Section 4)

### B.1  Highlight of Analysis Challenges

TCOM utilizes two techniques: communication arm set construction (§3.1.1) and observation-buffering broadcast (§3.1.2). Each technique alone can reduce the communication times to $O(\log T)$, and TCOM combines both to further reduce the communication times to $O(\log(\log T))$. This $O(\log(\log T))$ result comes directly from the TCOM policy design. The major challenge in analysis is to establish that—with these reduced and delayed sharing observations due to TCOM—the UCB-TCOM algorithm still preserves the near-optimal group and individual regrets. To do so, we show (1) the communication arm set construction technique can prevent communicating most optimal arm observations while allow communicating most suboptimal arm observations; (2) the observation sharing delay due to the observation-buffering broadcast technique (upon the communication arm set technique) does not have an intrinsic impact on UCB-TCOM's group and individual regret performance.

(1) Although the communication arm set construction technique shares the high level idea of ComEx (Madhushani & Leonard, 2021), the analysis of ComEx is not applicable to TCOM. Because the empirical means utilized for identifying the suboptimal arms in ComEx is only a special case of the tunable confidence intervals that TCOM relies on. Therefore, to show that agents can selectively share suboptimal arms' observations by the communication arm set technique, we prove two new results: (a) the optimal arm's observations are often not broadcast (i.e., the optimal arm is often not in the communication arm set, Theorem 2(ii)); (b) suboptimal arm observations are almost

always broadcast (i.e., the suboptimal arms are often in the set, Theorem 2(iii)). The (a) is proved via showing that the number of times that the optimal arm is in the communication arm set is no greater than that of an suboptimal arm (multiplied by a constant factor); thus is upper bounded by $O(\log T)$. This is based on revealing a relation between the bandit arm pull policy and the communication arm set construction. Then (b) is proved via showing whenever an agent pulls a suboptimal arm, the arm is inside the communication arm set with a high probability, and thus the suboptimal arm's observations are almost always buffered for later communication.

Our observation-buffering broadcast mechanism shares a high-level idea of doubling phase technique, but its specific design is different from known ones, and, consequently, its analysis addresses novel challenges. The doubling phase technique was utilized in CMA2B literature, e.g., Boursier & Perchet (2019); Wang et al. (2020b); Shi et al. (2021b), where their arm pull policies within a phase were either uniformly pulling each arm (Boursier & Perchet, 2019; Wang et al., 2020a) or sticking to several arms (Shi et al., 2021b). However, different from these previous "rigid" arm pull policies, UCB-TCOM applies the "flexible" UCB arm pull policy. In order to adapt the doubling phase technique to the UCB policy, we propose the observation-buffering broadcast mechanism which separately buffers the observation of each arms and respectively communicates each arm's compound observations whenever this arm's observation times is doubled (or generally, increased by a $\beta$ factor).

(2) Since this observation-buffering broadcast is different from known doubling phase algorithms, their existing analysis approaches are not applicable to observation-buffering. Hence, proving that this broadcast mechanism does not intrinsically deteriorate the near-optimal group and individual regrets of UCB-TCOM is a unique challenge. More specifically, we prove that the delay due to buffering only makes regrets worse by a constant factor. This is via showing that the observation-buffering mechanism separately buffers each arm's observations, and, therefore, the observation delay of each arm's observations (as well as the pulling times of this arm during the delay) can be respectively upper bounded by at most a constant factor multiplying this arm's total previous pulling times (see (9) of Theorem 1's proof as a formal expression).

## B.2 PRELIMINARIES

In this subsection, we provide several basic results and definitions that are crucial in the later analysis.

**Definition 1** (Type-I and Type-II decisions). *An agent makes a Type-I decision in time slot $t$ if in this time slot all arms' true reward means $\mu(k)$ lie inside this agent's tunable confidence intervals $(\texttt{tLCB}_t(k,\alpha), \texttt{tUCB}_t(k,\alpha))$ respectively. Otherwise, the agent makes a Type-II decision.*

**Lemma 1.** *For any arm $k \in [K]$, any agent $i \in [M]$, and time slot $t \in [T]$, the probability that the arm $k$'s true reward mean $\mu(k)$ lies inside its tunable confidence interval $(\texttt{tLCB}_t(k,\alpha), \texttt{tUCB}_t(k,\alpha))$ (calculated by agent $i$) is no less than $1 - 2t^{1-2\alpha^2}$.*

*Proof of Lemma 1.* We first bound the probability that the true reward mean is no less than the tunable upper confidence bound $\texttt{tUCB}$.

$$
\begin{aligned}
\mathbb{P}\left(\mu(k) \geqslant \texttt{tUCB}_t(k,\alpha)\right) &= \mathbb{P}\left(\mu(k) - \hat{\mu}_t(k) \geqslant \alpha\sqrt{\frac{\log t}{\hat{n}_t(k)}}\right) \\
&= \sum_{s=1}^{t} \mathbb{P}\left(\mu(k) - \hat{\mu}_t(k) \geqslant \alpha\sqrt{\frac{\log t}{\hat{n}_t(k)}} \,\middle|\, \hat{n}_t(k) = s\right)\mathbb{P}(\hat{n}_t(k) = s) \\
&\leqslant \sum_{s=1}^{t} \mathbb{P}\left(\mu(k) - \hat{\mu}_t(k) \geqslant \alpha\sqrt{\frac{\log t}{\hat{n}_t(k)}} \,\middle|\, \hat{n}_t(k) = s\right) \\
&\overset{(a)}{\leqslant} \sum_{s=1}^{t} t^{-2\alpha^2} \\
&\leqslant t^{1-2\alpha^2},
\end{aligned}
$$

where the inequality (a) is by the Hoeffding's inequality and that the random variable $\mu(k) - \hat{\mu}_t(k)$ is $\sqrt{1/4\hat{n}_t(k)}$-subgaussian. With a similar derivation, the probability that the true reward mean $\mu(k)$

is no greater than the tunable lower confidence bound $\texttt{tLCB}$, i.e., $\mu(k) \leqslant \texttt{tLCB}_t(k, \alpha)$, can also be upper bounded by $t^{1-2\alpha^2}$. Therefore, by excluding the above two events, the probability that the true reward mean lies inside the tunable confidence interval is no less than $1 - 2t^{1-2\alpha^2}$. $\qquad \square$

**Lemma 2** (Bound the number of times of Type-II decisions). *When $\alpha > 1$, the expected total number of times of Type-II decisions by any agent at any time slot in the system is finite. Specifically, it is less than $K\frac{2\alpha^2 - 1}{\alpha^2 - 1}$.*

*Proof of Lemma 2.* That a time slot $t$ is Type-II means that there exists at least one arm whose reward mean lies outside its tunable confidence interval. We bound the expected total number of times of Type-II decisions as follows,

$$
\sum_{t=1}^{T} \mathbb{P}\left(\exists k \in [K] \text{ such that } \mu(k) \notin (\texttt{tLCB}_t(k, \alpha), \texttt{tLCB}_t(k, \alpha))\right)
$$

$$
\leqslant \sum_{t=1}^{T} \sum_{k=1}^{K} \mathbb{P}\left(\mu(k) \notin (\texttt{tLCB}_t(k, \alpha), \texttt{tLCB}_t(k, \alpha))\right)
$$

$$
\leqslant K \sum_{t=1}^{T} \mathbb{P}\left(\mu(k) \notin (\texttt{tLCB}_t(k, \alpha), \texttt{tLCB}_t(k, \alpha))\right)
$$

$$
\overset{(a)}{\leqslant} 2K \sum_{t=1}^{T} t^{1-2\alpha^2}
$$

$$
\leqslant 2K \left(1 + \int_{t=1}^{T} t^{1-2\alpha^2} dt\right)
$$

$$
\overset{(b)}{\leqslant} MK \frac{2\alpha^2 - 1}{\alpha^2 - 1},
$$

where the inequality (a) holds for Lemma 1, and the inequality (b) holds because $\alpha > 1$ (i.e., $1 - 2\alpha^2 < -1$). $\qquad \square$

Next, we state a standard property of UCB algorithms and provide its proof for completeness.

**Lemma 3** (Adapted from (Yang et al., 2022, Lemma 2)). *If at any time $t \leqslant T$ agent $i \in [M]$ by $\texttt{UCB}$ makes a Type-I decision and pulls arm $k$, the total number of globally observed rewards of the arm $k$ by agent $i$ up to time $t$ is upper bounded as follows,*

$$
\hat{n}_t(k) \leqslant \frac{8 \log t}{\Delta_k^2}.
$$

*Proof of Lemma 3.* Given that the suboptimal arm $k$ is pulled by agent $i$, we have the following inequality

$$
2 \times \sqrt{\frac{2 \log t}{\hat{n}_t(k)}} \geqslant \Delta_k. \tag{8}
$$

Otherwise, we have

$$
\texttt{UCB}_t(1) = \hat{\mu}_t(1) + \sqrt{\frac{2 \log t}{\hat{n}_t(1)}}
$$

$$
\overset{(a)}{\geqslant} \mu(1) = \mu(k) + \Delta_k
$$

$$
\overset{(b)}{>} \mu(k) + 2\sqrt{\frac{2 \log t}{\hat{n}_t(k)}}
$$

$$
\overset{(c)}{>} \hat{\mu}_t(k) + \sqrt{\frac{2 \log t}{\hat{n}_t(k)}}
$$

$$
= \texttt{UCB}_t(k),
$$

which contradicts the pulling of arm $k$. Note that inequalities (a) and (c) are from Type-I decision's definition that reward means lie inside confidence interval, and inequality (b) is due to the inverse of (8). Therefore, (8) holds and rearranging (8) concludes the proof. □

### B.3 PROOFS OF GROUP REGRET (THEOREM 1

*Proof of Theorem 1.* To bound the regret, one needs to bound the number of times of pulling suboptimal arms by all $M$ agents in the system. Let $k$ be the index of a suboptimal arm. In UCB-TCOM, the reward observations of pulling arm $k$ are only broadcast (together as reward means) whenever arm $k$'s local observation times is increased by a $\beta$ ratio. So, there are delays before receiving other agents' observations. These delays may cause each agent pulling additional number of times of arm $k$. Next, we show that such delays do not significantly increase arm $k$'s overall pulling times in the system.

Note that the communication set contains all suboptimal arms when agents make Type-I decisions, in which case all suboptimal arm pulls increase the counter $N_t(k)$ by 1. Adding the other cost in Type-II decisions (at most $K\frac{2\alpha^2-1}{\alpha^2-1}$ by Lemma 2 and $\frac{2\alpha^2-1}{\alpha^2-1}$ for each arm), the total pulling times of an suboptimal arm $k$ among all $M$ agents are upper bounded by $\sum_{i\in[M]} N_t(k) + M\frac{2\alpha^2-1}{\alpha^2-1}$.

Recall that $\tau_t(k)$ denotes the last time slot (before time slot $t$) when agent $i$ broadcasts arm $k$'s buffered observations' average reward. To bound the first term in RHS, we have

$$\sum_{i\in[M]} N_t(k) \overset{(a)}{\leqslant} \beta \sum_{i\in[M]} N_{\tau_t(k)}(k) \overset{(b)}{\leqslant} \frac{8\beta \log T}{\Delta_k^2}, \tag{9}$$

where inequality (a) is because the counter $N_t(k)$ is at most $\beta$ times greater than its value at the last broadcast time slot, and inequality (b) is because $\sum_{i\in[M]} N_{\tau_t(k)}(k)$ is total global observations of arm $k$ at time slot $t$ which is less than $8\Delta_k^{-2} \log T$ by Lemma 3.

At last, we sum up the regret cost of pulling suboptimal arms in Type-I decisions and the other cost in Type-II decisions (in total at most $MK\frac{2\alpha^2-1}{\alpha^2-1}$ by Lemma 2 given $\alpha > 1$). So, the group regret is upper bounded as follows

$$\mathbb{E}[R_T(\mathcal{A})] \leqslant \sum_{k>1} \frac{8\beta \log T}{\Delta_k} + MK\frac{2\alpha^2-1}{\alpha^2-1}.$$

Notice that executing Algorithm 1, agents have the same arm pull behavior. So, the group regret are evenly divided into each agent's individual regret and thus the individual regret is upper bound is upper bounded as follows,

$$\mathbb{E}[R_T^{\text{ind}}(\mathcal{A})] \leqslant \sum_{k>1} \frac{8\beta \log T}{\Delta_k} + MK\frac{2\alpha^2-1}{\alpha^2-1}.$$

□

### B.4 PROOFS OF COMMUNICATION COSTS (THEOREM 2)

*Proof of Theorem 2(i).* No occurrence of communication for $\alpha \leqslant -\sqrt{2}$ is because that the pulled arm $A_t$ is never inside the communication set $\mathcal{C}_t(\alpha)$ in (3). When $\alpha \leqslant -\sqrt{2}$, an arm's tLCB is greater than its original UCB index, that is, $\text{tLCB}_t(k, \alpha) = \hat{\mu}_t(k) - \text{CI}_t(k, \alpha) \geqslant \hat{\mu}_t(k) + \sqrt{2\log t/\hat{n}_t(k)} = \text{UCB}_t(k)$; and its tUCB is smaller than its UCB index, that is, $\text{tUCB}_t(k, \alpha) = \hat{\mu}_t(k) + \text{CI}_t(k, \alpha) \leqslant \hat{\mu}_t(k) + \sqrt{2\log t/\hat{n}_t(k)} = \text{UCB}_t(k)$. Then, for the pulled arm $A_t$, the inverse of the communication set's condition holds:

$$\text{tLCB}_t(A_t, \alpha) \geqslant \text{UCB}_t(A_t)$$
$$\overset{(a)}{\geqslant} \text{UCB}_t(k), \forall k \neq A_t$$
$$\geqslant \text{tUCB}_t(k, \alpha), \forall k \neq A_t,$$

where the inequality $(a)$ is because that the pulled arm $A_t$ has the highest UCB index in time slot $t$. □

*Proof of Theorem 2(ii).* If a reward observation of optimal arm is broadcast by agent $i$ at time slot $t$, then there exists at least one suboptimal arm $k$ such that the following two events hold at the same time:

$$\mathtt{UCB}_t(1) \geqslant \mathtt{UCB}_t(k)$$
$$\mathtt{tLCB}_t(1, \alpha) < \mathtt{tUCB}_t(k, \alpha),$$

where the first event is from the $\mathtt{UCB}$ arm selection policy, and the second event holds because the pulled arm 1 belongs to the communication set. These two events imply:

$$\begin{cases} \hat{\mu}_t(1) + \sqrt{2 \log t / \hat{n}_t(1)} \geqslant \hat{\mu}_t(k) + \sqrt{2 \log t / \hat{n}_t(k)} \\ \hat{\mu}_t(1) - \mathtt{CI}_t(1, \alpha) < \hat{\mu}_t(k) + \mathtt{CI}_t(k, \alpha) \end{cases}$$
$$\implies \begin{cases} \hat{\mu}_t(1) - \hat{\mu}_t(k) \geqslant \sqrt{2 \log t / \hat{n}_t(1)} - \sqrt{2 \log t / \hat{n}_t(k)} \\ \hat{\mu}_t(1) - \hat{\mu}_t(k) < \alpha \left( \sqrt{\log t / \hat{n}_t(1)} + \sqrt{\log t / \hat{n}_t(k)} \right) \end{cases}$$
$$\implies (\sqrt{2} + \alpha) \sqrt{\log t / \hat{n}_t(1)} > (\sqrt{2} - \alpha) \sqrt{\log t / \hat{n}_t(k)}.$$

After rearranging the inequality, we show that the number of observations of the optimal arm 1 is upper bounded by a factor multiplying the number of observations of a suboptimal arm as follows,

$$\hat{n}_t(1) < \left( \frac{\sqrt{2} + \alpha}{\sqrt{2} - \alpha} \right)^2 \hat{n}_t(k). \tag{10}$$

Lemmas 2 and 3 together show that in the $\mathtt{UCB\text{-}TCOM}$ algorithm the maximum pulling times of any suboptimal arm $k$ is upper bounded as follows

$$\hat{n}_t(k) \leqslant \frac{8 \log T}{\Delta_k^2} + K \frac{2\alpha^2 - 1}{\alpha^2 - 1}. \tag{11}$$

Substituting the above inequality to (10)'s RHS and choosing $k = 2$ (because $1/\Delta_k^2$ is maximal when $k = 2$) for any time $t$, we have

$$\hat{n}_t(1) < \left( \frac{\sqrt{2} + \alpha}{\sqrt{2} - \alpha} \right)^2 \left( \frac{8 \log T}{\Delta_2^2} + K \frac{2\alpha^2 - 1}{\alpha^2 - 1} \right).$$

Notice that whenever the optimal arm is pulled and lies inside the communication set, the counter $\hat{n}_t(1)$ increases by 1. So, the buffering counter $N_t(1)$ is no greater than $\hat{n}_t(1)$. That is,

$$N_t(1) \leqslant \left( \frac{\sqrt{2} + \alpha}{\sqrt{2} - \alpha} \right)^2 \left( \frac{8 \log T}{\Delta_2^2} + K \frac{2\alpha^2 - 1}{\alpha^2 - 1} \right).$$

These $N_t(1)$ observations are aggregated and communicated at one time in each phase. These phases' lengths increases in a geometric sequence with common ratio $\beta$. So, the number of communicated messages of the optimal arm 1 is at most

$$\log_\beta(N_t(1)) \leqslant \log_\beta \left( \left( \frac{\sqrt{2} + \alpha}{\sqrt{2} - \alpha} \right)^2 \left( \frac{8 \log T}{\Delta_2^2} + K \frac{2\alpha^2 - 1}{\alpha^2 - 1} \right) \right) = O(\log(\log T)).$$

$\square$

*Proof of Theorem 2(iii).* We first show, if a suboptimal arm is pulled by a Type-I decision, its reward observation will definitely be broadcast, i.e., the pulled arm is inside the communication set. Because the pulled suboptimal arm $k$'s tunable lower confidence bound $\mathtt{tLCB}(k, \alpha)$ is less than the optimal arm 1's tunable upper confidence bound $\mathtt{tUCB}(1, \alpha)$: $\mathtt{tLCB}(k, \alpha) \leqslant \mu_k < \mu_1 \leqslant \mathtt{tUCB}(1, \alpha)$.

Lemma 2 shows that when $\alpha > 1$, agents make Type-I decisions in almost all time slots (except finite Type-II decisions). Therefore, almost all observations of suboptimal arms are broadcast. $\square$

*Proof of Theorem 2(iv).* We first show that when the optimal arm 1 is pulled and the agent makes a Type-I decision in time slot $t$, the event that $\texttt{tLCB}_t(1, \alpha) > \texttt{tLCB}_t(k, \alpha), \forall k \neq 1$ happens with small probability.

$$
\begin{aligned}
&\mathbb{P}\left(\texttt{tLCB}_t(1, \alpha) > \texttt{tLCB}_t(k, \alpha), \forall k \neq 1 | A_t = 1\right) \\
&= \mathbb{P}\left(\hat{\mu}(1) - \texttt{CI}_t(1, \alpha) > \hat{\mu}(k) + \texttt{CI}_t(k, \alpha), \forall k \neq 1 | A_t = 1\right) \\
&\leqslant \mathbb{P}\left(\hat{\mu}(1) > \texttt{CI}_t(k, \alpha), \forall k \neq 1 | A_t = 1\right) \\
&\leqslant \min_{k \neq 1} \mathbb{P}\left(\hat{\mu}(1) > \texttt{CI}_t(k, \alpha)\right) \\
&= \min_{k \neq 1} \mathbb{P}\left(\hat{\mu}(1) - \mu(1) > \texttt{CI}_t(k, \alpha) - \mu(1)\right) \\
&\overset{(a)}{\leqslant} \min_{k \neq 1} \mathbb{P}\left(\hat{\mu}(1) - \mu(1) > \alpha \frac{\Delta_k}{2\sqrt{2}} - \mu(1)\right) \\
&\overset{(b)}{\leqslant} \min_{k \neq 1} \exp\left(-(\alpha \Delta_k / 2 - \sqrt{2}\mu(1))\hat{n}_t(1)\right) \\
&= \exp\left(-(\alpha \Delta_2 / 2 - \sqrt{2}\mu(1))\hat{n}_t(1)\right),
\end{aligned}
\tag{12}
$$

where inequality (a) holds because (8) in Lemma 3's proof shows that when agent $i$ makes a Type-I decision to pull suboptimal arm $k$, the confidence interval $\texttt{CI}$'s width of this arm's reward mean is no less than half of the arm's reward gap, i.e., $\sqrt{\frac{2 \log t}{\hat{n}_t(k)}} \geqslant \frac{\Delta_k}{2}$, and inequality (b) holds because $\alpha \Delta_k - 2\sqrt{2}\mu(1) > 0$ and Hoeffding's inequality.

Next, we show that the number of times that the optimal arm 1's observations is not broadcast is finite.

$$
\begin{aligned}
&\mathbb{E}\left[\sum_{t \in [T]} \mathbb{1}\{A_t = 1 \text{ and the obs. is not broadcast}\}\right] \\
&\leqslant \mathbb{E}\left[\sum_{t \in [T]} \mathbb{1}\{\text{agent } i \text{ makes a Type-II decision in time } t\}\right] \\
&\quad + \mathbb{E}\left[\sum_{t \in [T]} \mathbb{1}\{A_t = 1 \text{ and the obs. is not broadcast and a Type-I decision is made in time } t\}\right] \\
&\leqslant \mathbb{E}\left[\sum_{t \in [T]} \mathbb{1}\{t \text{ is Type-II}\}\right] + \mathbb{E}\left[\sum_{t \in [T]} \mathbb{1}\{\texttt{tLCB}_t(1, \alpha) > \texttt{tUCB}_t(k, \alpha), \forall k \neq 1 | A_t = 1\}\right] \\
&\leqslant \mathbb{E}\left[\sum_{t \in [T]} \mathbb{1}\{t \text{ is Type-II}\}\right] + \mathbb{E}\left[\sum_{t \in [T]} \mathbb{1}\{\texttt{tLCB}_t(1, \alpha) > \texttt{tLCB}_t(k, \alpha), \forall k \neq 1 | A_t = 1\}\right] \\
&\overset{(a)}{\leqslant} \mathbb{E}\left[\sum_{t \in [T]} \mathbb{1}\{t \text{ is Type-II}\}\right] + \sum_{t \in [T]} \exp\left(-(\alpha \Delta_2 / 2 - \sqrt{2}\mu(1))\hat{n}_t(1)\right) \\
&\overset{(b)}{\leqslant} MK \frac{2\alpha^2 - 1}{\alpha^2 - 1} + \int_{n=0}^{T} \exp\left(-(\alpha \Delta_2 / 2 - \sqrt{2}\mu(1))n\right) dn \\
&\leqslant MK \frac{2\alpha^2 - 1}{\alpha^2 - 1} + \frac{2}{\alpha \Delta_2 - 2\sqrt{2}\mu(1)},
\end{aligned}
$$

where the inequality (a) is from applying (12) to the second term, and the inequality (b) bounds the number of times of Type-II decisions by Lemma 2. $\qquad \square$

## C APPLY PEELING TECHNIQUE TO RELAX THE TUNABLE PARAMETER'S CONSTRAINT

In this section, we apply peeling technique (Bubeck, 2010, §2.2) to relax Lemma 2 (thus Theorems 1 and 2(iii))'s $\alpha > 1$ constraint to $\alpha > \frac{1}{\sqrt{2}}$. This finer result helps to move the critical point 1 in Figure 2's $\alpha$-axis to $\frac{1}{\sqrt{2}}$, and extend the red arrow range (with near-optimal group regret and $O(\log(\log T))$ communications) to $\left( \frac{1}{\sqrt{2}}, \sqrt{2} \right)$.

**Lemma 4** (Bound the number of times of Type-II decisions via peeling technique). *Lemma 2's condition $\alpha > 1$ can be relaxed to $\alpha > \sqrt{\frac{1}{2\gamma}}$ via peeling technique for $\gamma \in (0,1)$. The total number of Type-II decisions is no greater than*

$$2MK \left( \frac{2\alpha^2\gamma + 1}{2\alpha^2\gamma - 1} + \frac{1}{\log(1/\gamma)(2\alpha^2\gamma - 1)^2} \right).$$

*Proof.* The key idea is applying a peeling argument to enhance Lemma 1's proof. Denote $\gamma \in (0,1)$. For any $s \leqslant t$, there exists $j \in \left\{ 0, 1, \ldots, \lceil \frac{\log t}{\log(1/\gamma)} \rceil \right\}$ such that $\gamma^j t < s \leqslant \gamma^{j+1} t$. Then, we have

$$
\begin{aligned}
\mathbb{P}\left(\mu(k) \geqslant \texttt{tUCB}_t(k, \alpha)\right) &= \mathbb{P}\left( \mu(k) - \hat{\mu}_t(k) \geqslant \alpha \sqrt{\frac{\log t}{\hat{n}_t(k)}} \right) \\
&\leqslant \mathbb{P}\left( \exists s \leqslant t : \mu(k) - \hat{\mu}_t(k) \geqslant \alpha \sqrt{\frac{\log t}{\hat{n}_t(k)}} \,\middle|\, \hat{n}_t(k) = s \right) \\
&\leqslant \sum_{j=0}^{\lceil \frac{\log t}{\log(1/\gamma)} \rceil} \mathbb{P}\left( \exists s \in \left(\gamma^j t, \gamma^{j+1} t\right] : \mu(k) - \hat{\mu}_t(k) \geqslant \alpha \sqrt{\frac{\log t}{\hat{n}_t(k)}} \,\middle|\, \hat{n}_t(k) = s \right) \\
&\leqslant \sum_{j=0}^{\lceil \frac{\log t}{\log(1/\gamma)} \rceil} \mathbb{P}\left( \exists s \in \left(\gamma^j t, \gamma^{j+1} t\right] : \mu(k) - \hat{\mu}_t(k) \geqslant \alpha \sqrt{\frac{\log t}{\gamma^j t}} \,\middle|\, \hat{n}_t(k) = s \right) \\
&\overset{(a)}{\leqslant} \sum_{j=0}^{\lceil \frac{\log t}{\log(1/\gamma)} \rceil} \exp\left( -\frac{\frac{\alpha^2 \log t}{\gamma^j t}}{2 \times (1/4\gamma^{j+1} t)} \right) \\
&\leqslant \sum_{j=0}^{\lceil \frac{\log t}{\log(1/\gamma)} \rceil} \exp\left( -2\alpha^2 \gamma \log t \right) \\
&\leqslant \left( 2 + \frac{\log t}{\log(1/\gamma)} \right) t^{-2\alpha^2\gamma},
\end{aligned}
$$

where the inequality (a) is from the maximal Hoeffding's inequality. Symmetrically, we also have

$$\mathbb{P}\left(\mu(k) \leqslant \texttt{tLCB}_t(k, \alpha)\right) \leqslant \left( 2 + \frac{\log t}{\log(1/\gamma)} \right) t^{-2\alpha^2\gamma}.$$

Together, we have

$$\mathbb{P}\left(\mu_k \notin (\texttt{tLCB}_t(k, \alpha), \texttt{tUCB}_t(k, \alpha))\right) \leqslant 2 \left( 2 + \frac{\log t}{\log(1/\gamma)} \right) t^{-2\alpha^2\gamma}.$$

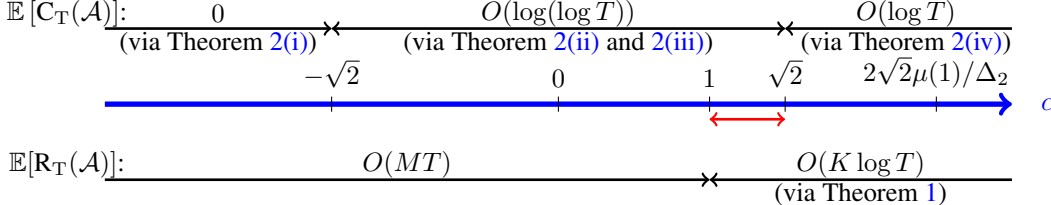

Figure 2: Impact of parameter $\alpha$ on UCB-TCOM's communication time $\mathbb{E}\left[C_\mathrm{T}(\mathcal{A})\right]$ and group regret $\mathbb{E}[\mathrm{R}_\mathrm{T}(\mathcal{A})]$. When $\alpha \in \left(1, \sqrt{2}\right)$, UCB-TCOM achieves the near-optimal group and individual regret upper bounds with $O(\log(\log T))$ communications.

Substituting Lemma 2's proof's third step with the above tighter inequality, we have

$$\sum_{t=1}^{T}\sum_{i=1}^{M}\sum_{k=1}^{K}\mathbb{P}\left(\mu_k \notin (\mathrm{tLCB}_t(k,\alpha), \mathrm{tUCB}_t(k,\alpha))\right)$$

$$\leqslant MK\sum_{t=1}^{T}\mathbb{P}\left(\mu_k \notin (\mathrm{tLCB}_t(k,\alpha), \mathrm{tUCB}_t(k,\alpha))\right)$$

$$\leqslant 2MK\sum_{t=1}^{T}\left(2 + \frac{\log t}{\log(1/\gamma)}\right)t^{-2\alpha^2\gamma}$$

$$\leqslant 2MK\left(1 + \int_{t=1}^{T}\left(2 + \frac{\log t}{\log(1/\gamma)}\right)t^{-2\alpha^2\gamma}dt\right)$$

$$\leqslant 2MK\left(\frac{2\alpha^2\gamma + 1}{2\alpha^2\gamma - 1} + \frac{1}{\log(1/\gamma)(2\alpha^2\gamma - 1)^2}\right),$$

where the last inequality requires the exist of integral, i.e., $2\alpha^2\gamma > 1$, from which one has $\alpha > \sqrt{\frac{1}{2\gamma}}$ and it becomes $\sqrt{\frac{1}{2}}$ when $\gamma \to 1$. $\qquad\square$

## D  TUNABILITY OF TCOM

### D.1  ALGORITHM DESIGN DISCUSSION

The parameter $\alpha$ tunes the aggressiveness of identifying an arm as suboptimal. When the condition is aggressive ($\alpha$ is small), the communication arm set $\mathcal{C}_t^{(i)}(\alpha)$ is small: most arms inside $\mathcal{C}_t^{(i)}(\alpha)$ are suboptimal with high probability, but there may exist some suboptimal arms outside $\mathcal{C}_t^{(i)}(\alpha)$, i.e., $\mathcal{C}_t^{(i)}(\alpha)$ fails to cover a subset of suboptimal arms. Therefore, the observations of those suboptimal arms outside the communication arm set cannot be shared, which causes more explorations on them. That is, the cooperative algorithm's regret may be large when a communication arm set with small $\alpha$. When the condition is conservative ($\alpha$ is large), $\mathcal{C}_t^{(i)}(\alpha)$ tends to be large: it contains not only most suboptimal arms but sometimes the optimal arm as well. As the observation times of optimal arm are also broadcast, the communication times would be large when $\alpha$ is large.

The parameter $\beta$ controls the frequency of communicating an arm's observation. When $\beta$ is large, communication times reduce at the expense of longer delays before other agents receive the observations. Therefore, the group regret increases. And vice versa when $\beta$ decreases.

### D.2  THEORETICAL DISCUSSION

**Impact of communication parameter $\alpha$.**  (1) Theorem 2(i) shows that when $\alpha \leqslant -\sqrt{2}$, there is no communication among agents and thus there are no global observations for agents to optimize their decisions.  (2) Theorems 2(iii) and 2(iv) shows that when $\alpha > \frac{2\sqrt{2}\mu(1)}{\Delta_2}(> \sqrt{2})$ almost all

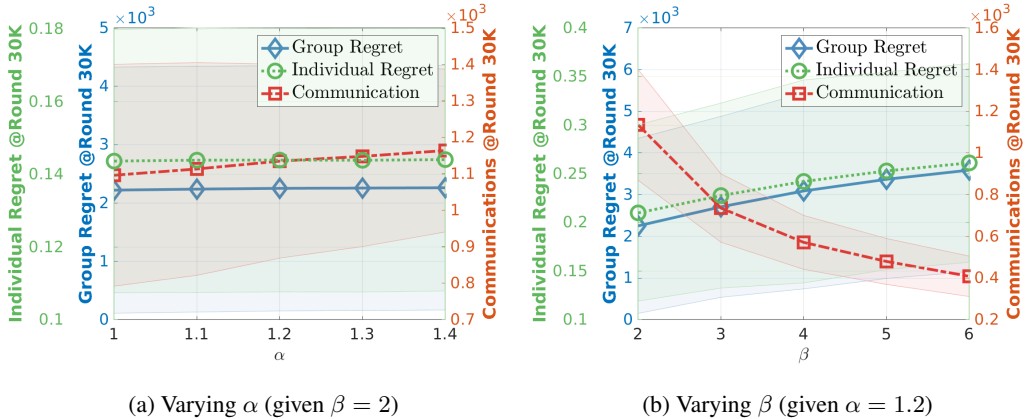

(a) Varying $\alpha$ (given $\beta = 2$)

(b) Varying $\beta$ (given $\alpha = 1.2$)

Figure 3: Impact of communication set parameter $\alpha$ with fixed $\beta = 2$ in Figures 3a; and buffering ratio $\beta$ with fixed $\alpha = 1.2$ in Figures 3b

reward observations are broadcast. Hence the number of global observations that were broadcast is similar to that of total observations of agents. (3) Theorem 2(ii) shows that when $-\sqrt{2} < \alpha < \sqrt{2}$, the number of broadcasts of the optimal arm's observations is $O(\log(\log T))$. Additionally, the observation buffering mechanism also aggregates the suboptimal arms $O(\log T)$ observations into $O(\log(\log T))$ broadcasts. So, in this case, UCB-TCOM only incurs $O(\log(\log T))$ communication time. (4) Combined with (3), Theorem 1 shows that when $1 < \alpha < \sqrt{2}$, UCB-TCOM has near-optimal group and individual regret upper bounds with only $O(\log(\log T))$ communication cost. We refer to the interval $\left(1, \sqrt{2}\right)$ as *the preferable range* for $\alpha$. Figure 2 summarizes the impact of parameter $\alpha$ on the regret-communication trade-off.

**Impact of observation buffering ratio $\beta$.** In addition to $\alpha$, the $\beta$ can also be used to tune the regret-communication trade-off. When the $\beta$ increases, Theorems 1 shows that both group and individual regrets increase. At the same time, note that $\beta$ is the logarithm base in (6) of Theorem 2(ii), which means the communication times of TCOM decreases as $\beta$ increases. Similarly, the regrets decrease and communication times increases as the ratio $\beta$ decreases.

### D.3 SIMULATIONS ON TUNABILITY

Figure 3 illustrates the impact of communication set parameter $\alpha$ and observation buffering ratio $\beta$ on the performance of UCB-TCOM. From regret aspect, Figure 3a shows that both the group and individual regrets do not change much as $\alpha$ increases, which confirms the regret upper bounds in Theorem 1 where $\alpha$ does not appear on the dominating logarithmic term of (4) and (5). Figure 3b shows that the regrets increase with respect to $\beta$, which corresponds to the appearance of $\beta$ in the dominating logarithmic term of regret bounds in Theorem 1. From communication aspect, Figure 3a and Figure 3b also corroborate (6) of Theorem 2, where the communication times increases with respect to $\alpha$ and decreases with respect to $\beta$.

### E  SIMULATIONS ON THE PERFORMANCE OF UCB-TCOM IN DIFFERENT ENVIRONMENTS

In Figures 4a and 4b, we vary the number of agents $M$ in $\{5, 25, 45, 65, 85, 105\}$ and the number of arms $K$ in $\{10, 20, 40, 60, 80, 100\}$ respectively while fixing other default values. The group regret in Figure 4a is flat (compared to Figure 4b's group regret), corroborating that it does not increase rapidly with respect to number of agents $M$ (discussed in Section 4.2(**b**), also note that the second term in (4) does not depend on $T$. Its decreasing individual regret curve corroborates the scalability advantage of the multi-agent system — the more agents in the system, the smaller the maximum individual cost these agents need to pay (discussed in Section 4.2(**c**)). Figure 4b shows that both

---

**Algorithm 2** The `TS-UCB` Algorithm (for each agent)

---

1: **Input:** the communication arm set parameter $\alpha$ and buffering ratio $\beta$
2: **Initialization:** $\hat{n}_t(k) = 0, N_t(k) = 0, \hat{\mu}_t(k) = 0, \tau_t(k) = 0$
3: **for** each decision round $t$ **do**
4:     For each arm $k \in \mathcal{K}$, sample $\theta_t(k)$ from the Beta$(\hat{n}_t(k)\hat{\mu}_t(k) + 1, \hat{n}_t(k) + 1)$ distribution.
5:     Pull arm $A_t$ with the highest $\theta_t(k)$
6:     Observe arm $A_t$'s reward $X_t(A_t)$
7:     **if** $A_t \in \mathcal{C}_t(\alpha)$ **then**
8:         Increase $N_t(A_t)$ by 1
9:         Update this phase's empirical mean $\tilde{\mu}_t(A_t)$
10:    **end if**
11:    **if** $N_t(A_t) \geqslant \lceil \beta N_{\tau_t(A_t)}(A_t) \rceil$ **then**
12:        Broadcast the message $(\tilde{\mu}_t(A_t), N_t(A_t), A_t)$
13:        $\tau_t(A_t) \leftarrow t$
14:    **end if**
15: **end for**
16: **for** each newly received message $(\tilde{\mu}_t(k), N_t(k), k)$ from the past decision round **do**
17:    Update the empirical mean $\hat{\mu}_t(k) \leftarrow \frac{\hat{\mu}_t(k)\hat{n}_t(k) + \tilde{\mu}_t(k)\lfloor N_t(k)(1-1/\beta) \rfloor}{\hat{n}_t(k) + \lfloor N_t(k)(1-1/\beta) \rfloor}$
18:    Increase $\hat{n}_t(k)$ by $\lfloor N_t(k)(1 - 1/\beta) \rfloor$
19:    Update the communication arm set $\mathcal{C}_t(\alpha)$ via (3) based on tunable confidence bounds
20: **end for**

---

group and individual regrets, and communication cost increase linearly as the number of arms $K$ increases, which signifies the linear dependence of regret on the number of arms.

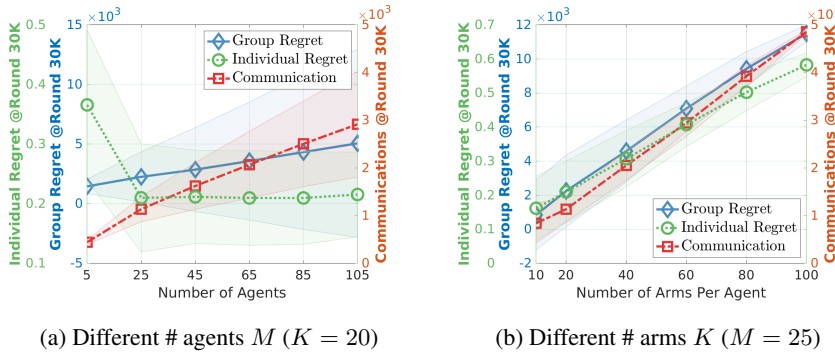

(a) Different # agents $M$ ($K = 20$)        (b) Different # arms $K$ ($M = 25$)

Figure 4: The performance of `UCB-TCOM` with different number of agents and arms

## F APPLY TCOM TO OTHER BANDITS ALGORITHMS

In this section, we illustrate how `TCOM` can be applied to other bandits algorithms. Specifically, we devise the `TS-TCOM` algorithm based on the Thompson sampling (`TS`) algorithm (Agrawal & Goyal, 2012) and the `AAE-TCOM` algorithm based on the active arm elimination (`AAE`) algorithm (Even-Dar et al., 2006).

For the ease of presenting `TS-TCOM`, we restrict our reward distributions to Bernoulli (same as our simulation setting) and assume the prior of all arms' rewards are Beta$(1, 1)$. Then, at time $t$, the arm $k$'s reward posterior distribution is Beta$(\hat{n}_t(k)\hat{\mu}_t(k) + 1, \hat{n}_t(k) + 1)$, where the $\hat{n}_t(k)$ and $\hat{\mu}_t(k)$ are the *global* pulling times and the *global* reward mean estimate of arm $k$ respectively. We present `TS-TCOM` in Algorithm 2. The only difference between `TS-TCOM` and `UCB-TCOM` is in Line 4-5.

In `AAE-TCOM`, agents need to construct a *global* candidate arm set $\mathcal{D}_t$ which is defined as follows,

$$\mathcal{D}_t := \{k \in [K] : \text{UCB}_t(k) \geqslant \text{LCB}_t(k'), \forall k' \in [K]\}, \tag{13}$$

---

**Algorithm 3** The `AAE-TCOM` Algorithm (for each agent)

---

1: **Input:** the communication arm set parameter $\alpha$ and buffering ratio $\beta$
2: **Initialization:** $\hat{n}_t(k) = 0, N_t(k) = 0, \hat{\mu}_t(k) = 0, \tau_t(k) = 0$
3: **for** each decision round $t$ **do**
4:     Update candidate arm set $\mathcal{D}_t$ via (13)
5:     Pull arm $A_t$ with the smallest *global* observation times $\hat{n}_t(k)$ among arms in $\mathcal{D}_t$
6:     Observe arm $A_t$'s reward $X_t(A_t)$
7:     **if** $A_t \in \mathcal{C}_t(\alpha)$ **then**
8:         Increase $N_t(A_t)$ by 1
9:         Update this phase's empirical mean $\tilde{\mu}_t(A_t)$
10:     **end if**
11:     **if** $N_t(A_t) \geqslant \lceil \beta N_{\tau_t(A_t)}(A_t) \rceil$ **then**
12:         Broadcast the message $(\tilde{\mu}_t(A_t), N_t(A_t), A_t)$
13:         $\tau_t(A_t) \leftarrow t$
14:     **end if**
15: **end for**
16: **for** each newly received message $(\tilde{\mu}_t(k), N_t(k), k)$ from the past decision round **do**
17:     Update the empirical mean $\hat{\mu}_t(k) \leftarrow \frac{\hat{\mu}_t(k)\hat{n}_t(k) + \tilde{\mu}_t(k)\lfloor N_t(k)(1-1/\beta)\rfloor}{\hat{n}_t(k) + \lfloor N_t(k)(1-1/\beta)\rfloor}$
18:     Increase $\hat{n}_t(k)$ by $\lfloor N_t(k)(1 - 1/\beta) \rfloor$
19:     Update the communication arm set $\mathcal{C}_t(\alpha)$ via (3) based on tunable confidence bounds
20: **end for**

---

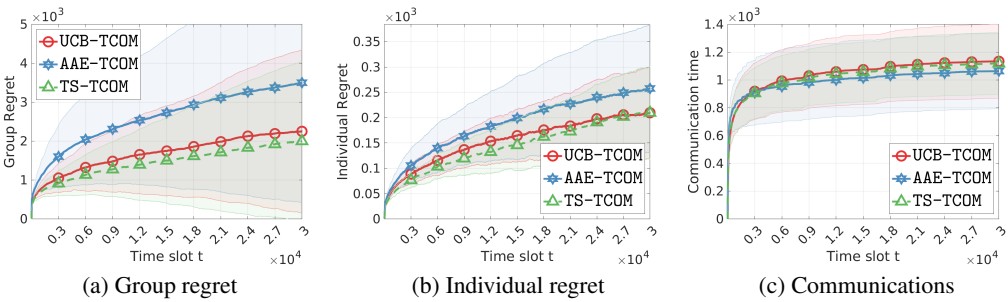

(a) Group regret        (b) Individual regret        (c) Communications

Figure 5: `UCB-TCOM` vs. `AAE-TCOM`, `TS-TCOM`

where $\text{UCB}_t(k) := \hat{\mu}_t(k) + \sqrt{2\log t / \hat{n}_t(k)}$ is the same as `UCB-TCOM`, and $\text{LCB}_t(k) := \hat{\mu}_t(k) - \sqrt{2\log t / \hat{n}_t(k)}$. We note that since both $\hat{\mu}_t(k)$ and $\hat{n}_t(k)$ are global, all agents' candidate arm set are the same. Then, at each time slot $t$, agents pull the arm with smallest observation times among all arms in the candidate arm set, i.e., $\arg\min_{k \in \mathcal{D}_t} \hat{n}_t(k)$. We present `AAE-TCOM` in Algorithm 3. The difference between `AAE-TCOM` and `UCB-TCOM` is also in Line 4-5.

In Figure 5, we report the performance comparison between `UCB-TCOM`, `TS-TCOM`, and `AAE-TCOM`. The simulations are conducted under the default setting in §5. In Figure 5a and 5b, the `AAE-TCOM` algorithm has the worse regret performance than `UCB-TCOM`, and `UCB-TCOM` is worse than `TS-TCOM`. This matches the folklore in bandits that the empirical performance of `TS` is better than that of `UCB`, and `UCB` is better than `AAE`. Comparing the individual regret in Figure 5b to the group regret in Figure 5a shows that the `AAE-TCOM` and `TS-TCOM` algorithms have good individual regret performance. Figure 5c shows that these three algorithms need the similar communication times, which validates the communication efficiency of `TS-TCOM` and `AAE-TCOM`.

We believe that the `TS-TCOM` and `AAE-TCOM` algorithms (like `UCB-TCOM`) also enjoy the $O(\log(\log T))$ communication times and near-optimal group and individual regrets. However, this requires new analysis, especially for the Thompson sampling case, because its Bayesian approach is very different from the method utilized in `UCB` and `TS-TCOM` needs to validate the symmetric learning structure. Studying both new algorithms' theoretical performance is an interesting future work.

# G  EXTENSION OF TCOM TO CMA2B WITH COMMUNICATION WITH DELAYS AND COMMUNICATION OVER A GRAPH

In this section, we extend the basic CMA2B model in §2 to allow deterministic communication delays and also from that the agents can broadcast to any others to the case that agents residing on a network can only communicate with their neighbors (i.e., cannot broadcast). In the communication network model, we also allow agents to passing its received messages to other neighbors, i.e., allow message-passing. We show that UCB-TCOM (with minor changes) still enjoys the near-optimal group and individual regret upper bounds as in Theorem 1 with only constant additional cost (independent of $T$). We also note that the communication times results in Theorem 2 still hold because its proofs are not influenced by these extensions.

## G.1  DETERMINISTIC DELAY

Denote $d \in \mathbb{N}^+$ as the deterministic delay of agents' message broadcasts. The deterministic delay can also be relaxed to random delay but with the $d$ as the delay upper bound. Algorithmically, the global UCB utilized for selecting arms to pull should be re-defined to those observations that have been broadcast $d$ *time slots before*. Because only these reward observations have been received by all agents even with the communication delay and thus agents utilize the same observations to make decisions.

**Theorem 3.** *When the communication arm set parameter $\alpha > 1$ and buffering-ratio $\beta > 1$, given all delays of communication is no greater than $d$,* UCB-TCOM *attains a near-optimal group regret upper bound in terms of number of decision rounds $T$, arms $K$, and agents $M$, or formally,*

$$\mathbb{E}[\mathbf{R}_{\mathrm{T}}(\mathcal{A})] \leqslant \sum_{k>1} \frac{8\beta \log T}{\Delta_k} + MK\frac{2\alpha^2 - 1}{\alpha^2 - 1} + dM\sum_{k>1}\Delta_k$$

*and* UCB-TCOM *also attains a near-optimal individual regret upper bound, or formally,*

$$\mathbb{E}[\mathbf{R}_{\mathrm{T}}^{\mathrm{ind}}(\mathcal{A})] \leqslant \sum_{k>1} \frac{8\beta \log T}{M\Delta_k} + K\frac{2\alpha^2 - 1}{\alpha^2 - 1} + d\sum_{k>1}\Delta_k.$$

*Proof of Theorem 3.* The proof procedure of Theorem 1 still applies to Theorem 3. The only change is that the inequality (9) should be updated as follows,

$$\sum_{i\in[M]} N_t^i(k) \overset{(a)}{\leqslant} \sum_{i\in[M]} (\beta N_{\tau_t^i(k)}^i(k) + d) \leqslant \beta \sum_{i\in[M]} N_{\tau_t(k)}(k) + dM \overset{(b)}{\leqslant} \frac{8\beta \log T}{\Delta_k^2} + dM,$$

where inequality (a) is because the counter $N_t(k)$ is at most $\beta$ times greater than its value at the last broadcast time slot plus the delay $d$, and inequality (b) is because $\sum_{i\in[M]} N_{\tau_t(k)}(k)$ is total global observations of arm $k$ at time slot $t$ which is less than $8\Delta_k^{-2} \log T$ by Lemma 3.

With the rest proof the same as Theorem 1's, UCB-TCOM's group regret is upper bounded as follows:

$$\mathbb{E}[\mathbf{R}_{\mathrm{T}}(\mathcal{A})] \leqslant \sum_{k>1} \frac{8\beta \log T}{\Delta_k} + dM\sum_{k>1}\Delta_k + MK\frac{2\alpha^2 - 1}{\alpha^2 - 1}.$$

As the symmetry of UCB-TCOM still holds, the individual regret upper bound immediately follows,

$$\mathbb{E}[\mathbf{R}_{\mathrm{T}}^{\mathrm{ind}}(\mathcal{A})] \leqslant \sum_{k>1} \frac{8\beta \log T}{M\Delta_k} + d\sum_{k>1}\Delta_k + K\frac{2\alpha^2 - 1}{\alpha^2 - 1}.$$

$\square$

## G.2  COMMUNICATION TOPOLOGY: PEER-TO-PEER AND MESSAGE PASSING COMMUNICATION

We assume the communication network is a connected graph, i.e., there exists a path between any two nodes, and the diameter of the graph is $D$. In the message-passing protocol (following Dubey

et al. (2020)'s model), agents $i \in [M]$ communication via messages $(i, t, \tilde{\mu}_t^i(A_t^i), N_t^i(A_t^i), A_t^i)$. This message is sent to its neighbors in the graph and then is forwarded to by any agents when receives it to their neighbors until the time slot $t + D$. Under this protocol, the broadcast message will final reach to any agents with a delay at most $D$. In algorithmic aspect, the TCOM algorithm still works with minor modifications:

1. Change the Line 11 in Algorithm 1 from broadcast to communication to neighbors;
2. Add a line after Line 11 that an agent should also send all its received messages with time index $t' < t + D$ to neighbors on the graph;
3. Line 15 should be changed to receive all messages that have not been received before (i.e., with unique $i, t$ prefix in the message).

With the above modifications in UCB-TCOM, the algorithm solves a problem equivalent to a CMA2B with communication delay $D$ on a complete graph. Therefore, based on Theorem 3, one obtains the following corollary to bound the group and individual regrets of the modified UCB-TCOM algorithm.

**Corollary 4.** *When the communication arm set parameter $\alpha > 1$ and buffering-ratio $\beta > 1$, given all agents residing on a network with diameter $D$, UCB-TCOM attains a near-optimal group regret upper bound in terms of number of decision rounds $T$, arms $K$, and agents $M$, or formally,*

$$\mathbb{E}[\mathbf{R}_T(\mathcal{A})] \leqslant \sum_{k>1} \frac{8\beta \log T}{\Delta_k} + MK \frac{2\alpha^2 - 1}{\alpha^2 - 1} + DM \sum_{k>1} \Delta_k$$

*and UCB-TCOM also attains a near-optimal individual regret upper bound, or formally,*

$$\mathbb{E}[\mathbf{R}_T^{\text{ind}}(\mathcal{A})] \leqslant \sum_{k>1} \frac{8\beta \log T}{M\Delta_k} + K \frac{2\alpha^2 - 1}{\alpha^2 - 1} + D \sum_{k>1} \Delta_k.$$

