# OpenReview forum: "Achieving Near-Optimal Individual Regret & Low Communications in Multi-Agent Bandits"
_ICLR.cc/2023/Conference — ICLR 2023 poster_

### Official Review · Reviewer_VQR1 · 2022-10-19

**Confidence:** 4
**Correctness:** 3
**Technical Novelty And Significance:** 3
**Empirical Novelty And Significance:** 3
**Recommendation:** 6

**Clarity, Quality, Novelty And Reproducibility:**

Despite some weaknesses, the paper is very well written.
The idea used in TCOM is well known: the doubling trick. It has already been used for minimizing communication rounds (Hillel et al 2014). However, the present paper introduces some novelty that are sound.


**Strength And Weaknesses:**

The proposed communication policy is sound. The analysis and the experiments are well done.
However, the paper presents some shortcomings.

1/The definition of both regrets seems incorrect. Indeed, there is an expectation in the regret, while there is no random variable inside: \mu(A^i_t) is the mean reward of arm chosen by agent i at time t. Did the authors take the expectation over the choices of players? Why not using the usual definition of pseudo-regret (cumulative difference between the expected reward of the optimal arm and the expected reward of the chosen arm)?

2/ In comparison to DEP2, UCB-TCOM is outperformed in two metrics (group regret and communication time). So the only advantage of UCB-TCOM is that its individual regret is smaller. So in the comparison, individual regret is crucial, and it has to be better motivated. May be by enhancing fairness ?

Minor comments:
Table 1 presents an error: the communication time is in O(1) for DEP2.
The authors could refer some works done in the collaborative exploration setting. Indeed, explore then commit algorithm can also reach a near-optimal regret (Garivier et al 2016), and despite different contexts, the two following papers use similar ideas: doubling trick for communication policy in (Hillel et al 2016), communicating only about bad arms in (Feraud et al 2019).

(Hillel et al 2013) Distributed exploration in multi-armed bandits, NeurIPS 2016.

(Feraud et al 2019) Decentralized exploration in multi-armed bandits, ICML 2019.

(Garivier et al 2016), On explore the commit strategies, NeurIPS 2016.


**Summary Of The Paper:**

This paper study the collaborative MAB, where agents can communicate together to handle the same MAB problem. The authors propose a new notion of regret: the individual regret is the maximum regret over the players. This notion of regret is given in addition to the usual notion of regret. This regret could be related to the notion of fairness, since the closer the individual regret to regret the greater the fairness is.
In collaborative MAB the goal is to minimize the regret but also the communication cost. They provide a communication policy that has a communication cost in O(log log T) and that benefits from a near-optimal regret bound for both notions of regret. In some experiments, they show that their algorithm (UCB-TCOM) provides a good compromise between group regret and individual regret, while they obtain a low communication cost, even if it is not the smallest.


**Summary Of The Review:**

Overall, despite some shortcomings that can be fixed, this paper is technically sound, well written, and presents some novelties.
_____________________________
I read the rebuttal and I thank the authors for their answers.

---

> ### Author Response · Authors · 2022-11-10
> **On the Related Works about Collaborative Exploration**
>
> - *"The authors could refer some works done in the collaborative exploration setting."*
>
> We thank the reviewer for providing this line of related works. We enhanced the comparison to the collaborative exploration case in Appendix A in the revised submission. We quote it as follows:
>
> >Cooperative pure exploration (best arm identification)) (Hillel et al., 2013;
> Tao et al., 2019; Féraud et al., 2019; Karpov et al., 2020)
> study how multi-agent cooperation can reduce the sample complexity of identifying the best arm which is a different objective from our work on regret minimization.
> Some of their high-level ideas in communication are similar to $\texttt{TCOM}$, e.g., Hillel et al. (2013) utilize the doubling phase for communication and  Féraud et al. (2019)’s algorithm communicates bad arms.
> However, as we discussed in Appendix B.1, our algorithm is novel in specific design detail, and thus introduces unique challenges that cannot be addressed by the techniques of these known works.

---

> ### Author Response · Authors · 2022-11-10
> **On the Minor Comment about Table 1**
>
> - *"Minor comments: Table 1  presents an error: the communication time is in O(1) for DEP2."*
>
> Table 1 says that the communication times of $\texttt{DPE2}$ is $O(K^2 M^2)$, which follows (Wang et al., 2020a, Theorem 2). If both the number of arms $K$ and the number of agents $M$ are constant, it would reduce to $O(1)$.

---

> ### Author Response · Authors · 2022-11-10
> **On Comparison to DPE2 and Motivating Fairness**
>
> - *"2/ In comparison to DEP2, UCB-TCOM is outperformed in two metrics (group regret and communication time). So the only advantage of UCB-TCOM is that its individual regret is smaller. So in the comparison, individual regret is crucial, and it has to be better motivated. May be by enhancing fairness ?"*
>
> We highly appreciate this reviewer's comments.
> In this paper, we provided two typical examples---drown swarms and inference engine in network management, in which minimizing individual regret is a much more important objective than group regret and communication times.
> Hence, the $\texttt{UCB-TCOM}$ algorithm with the near-optimal individual regret can efficiently address these applications, while $\texttt{DPE2}$ with the inferior individual regret performance cannot.
>
> In this rebuttal submission's introduction, besides the existed examples of drone swarms, distributed systems, and computer networks for motivating the individual regret metric,
> we added the max-min fairness metric (and its applications) as another example to motivate the importance of studying the individual regret for achieving fair performance.
> We quote the text as follows,
>
> >For another thing, in network optimization literature, the max-min fairness metric---maximize the minimal individual reward---is widely used to measure a system's fairness (Srikant & Ying, 2013, $\S2.2.1$), such as fair queuing (Demers et al., 1989). Since the regret is the opposite of reward, optimizing the max-min fairness is also equivalent to minimizing the bottleneck agent's regret. Other fairness motivation examples can be found in political philosophy (Rawls, 2004).

---

> ### Author Response · Authors · 2022-11-10
> **On Regret Definitions**
>
> We highly appreciate the reviewer's instructive comments. Next, we reponse these comments point-by-point.
>
> - *"1/The definition of both regrets seems incorrect. Indeed, there is an expectation in the regret, while there is no random variable inside: \mu(A^i_t) is the mean reward of arm chosen by agent i at time t. Did the authors take the expectation over the choices of players? Why not using the usual definition of pseudo-regret (cumulative difference between the expected reward of the optimal arm and the expected reward of the chosen arm)?"*
>
> Thanks for pointing out this issue. We replaced the $\mu(A_t^{(i)})$ with $X_t^{(i)}(A_t^{(i)})$ in both group and individual regret definitions ((1) and (2)) in the revision submission.
> We note that since $\mathbb{E}[X_t^{(i)}(A_t^{(i)})] = \mathbb{E}[\mu(A_t^{(i)})]$, this revision has no impact on the result correctness in this paper.

---

### Official Review · Reviewer_coEu · 2022-10-25

**Confidence:** 3
**Correctness:** 4
**Technical Novelty And Significance:** 3
**Empirical Novelty And Significance:** Not applicable
**Recommendation:** 6

**Clarity, Quality, Novelty And Reproducibility:**

Typo: In line 292, the lower bound is on group regret (not individual); the superscript 'ind' should be removed I reckon?

Is factor of M the best possible improvement? In particular, how does group as well as individual regret compare with lower bounds for the scenario where an entire vector of M reward realizations is broadcasted per period? This, in my opinion, is a more interesting statistical baseline compared to the no communication one.

Can you black-box the algorithm into communication and learning modules while preserving regret guarantees? This begs the question, is communication efficiency a fundamental property of the algorithm itself? Can you comment on whether Thompson Sampling might work using the same broadcast signals and the same broadcast rate?

Is it possible to trace out a Pareto frontier of regret (individual/group) vs. communication overhead?

**Strength And Weaknesses:**

The main contribution appears to be showing that a factor of M improvement in group as well as individual regret is possible using only O(log log T) rounds of (broadcast) communications between the M agents. The improvement is relative to the scenario where the agents do not communicate and simply keep playing their own instance, oblivious to the rewards accrued by others. The communication efficiency of O(log log T) is achieved by sharing information about (empirically) sub-optimal arms as opposed to optimal ones, which I find to be a very neat idea.



**Summary Of The Paper:**

This work focuses on a cooperating learning version of the classical multi-armed bandit problem, where M agents play the same problem instance over T rounds. Agents can communicate their observations through global broadcasts. The authors propose an algorithm that needs O(log log T) communication rounds to achieve logarithmic individual as well as group regret.

**Summary Of The Review:**

The paper is generally well written and a pleasure to read. Appears to be technically sound, comprehensive in literature survey, and meaningful in contribution as it improves upon extant baselines by a considerable margin. It would be great if the authors could address or at least remark upon the questions raised earlier; that would really cement the contributions well and make it accessible for a more general audience. Overall, I vote for an acceptance.

---

> ### Author Response · Authors · 2022-11-10
> **On Tracing out Pareto Frontier**
>
> - *"Is it possible to trace out a Pareto frontier of regret (individual/group) vs. communication overhead?"*
>
> We believe this is possible but requires substantial efforts upon our current work.
> We added a comment in the conclusion section and placed it as an interesting future work in the revision submission:
>
> >We note that $\texttt{TCOM}$ can also be tuned to balance the regret and communications. One interesting future work is to systemically study the Pareto frontier of group/individual regrets vs. communication times from the perspective of the upper bounds of $\texttt{TCOM}$ and the intrinsic lower bounds (difficulties) of $\texttt{CMA2B}$ as well.

---

> > ### Comment · Reviewer_coEu · 2022-11-24
> > **Post author-rebuttal**
> >
> > I thank the authors for providing a detailed response. I appreciate the nuances the authors highlight, and would like to see perhaps a concise discussion on these aspects incorporated into the revision. I will keep my current score.

---

> ### Author Response · Authors · 2022-11-10
> **On Applying TCOM to Thompson Sampling**
>
> - *"Can you black-box the algorithm into communication and learning modules while preserving regret guarantees? This begs the question, is communication efficiency a fundamental property of the algorithm itself? Can you comment on whether Thompson Sampling might work using the same broadcast signals and the same broadcast rate?"*
>
> We can apply the $\texttt{TCOM}$ policy to other bandits algorithms, e.g., Thompson sampling ($\texttt{TS}$) and active arm elimination ($\texttt{AAE}$), and obtain the $\texttt{TS-TCOM}$ and $\texttt{AAE-TCOM}$ algorithms.
> In this revision submission, we provided the pseudo-codes and simulations of $\texttt{AAE-TCOM}$ and $\texttt{TS-TCOM}$ in Appendix E (we also added a footnote in the main text pointing to this).
> The simulations show that these two algorithms have similar performance to $\texttt{UCB-TCOM}$ in the main paper.
>
> We believe that the $\texttt{TS-TCOM}$ and $\texttt{AAE-TCOM}$ algorithms (like $\texttt{UCB-TCOM}$) also enjoy the $O(\log (\log T))$ communication times and near-optimal group and individual regrets.
> However, this requires new analysis, especially for the Thompson sampling case, because its Bayesian approach is very different from the method utilized in $\texttt{UCB}$,
> and $\texttt{TS-TCOM}$ needs to validate the symmetric learning structure. Studying both new algorithms' theoretical performance is an interesting future work.

---

> ### Author Response · Authors · 2022-11-10
> **On the Optimality of Lower Bound**
>
> - *"Is factor of M the best possible improvement? In particular, how does group as well as individual regret compare with lower bounds for the scenario where an entire vector of M reward realizations is broadcasted per period? This, in my opinion, is a more interesting statistical baseline compared to the no communication one."*
>
> Yes, the factor $M$ is indeed the best improvement.
> The lower bounds in (a) of $\S4.2$ hold for the case that *"an entire vector of M reward realizations is broadcasted per period"*.
> Because they are from the centralized (single agent) bandits which are equivalent to that multiple agents can freely communicate with each other.
>
> We revised a sentence of (a) in $\S4.2$ to clarify this:
>
> >Since the $\texttt{CMA2B}$ model has the same objective
>         as the centralized (single agent) MAB model,
>         it inherits this group regret lower bound (Wang et al., 2020a, $\S1.2$) under any possible communication policies.

---

> ### Author Response · Authors · 2022-11-10
> **On the Superscript Typo**
>
> We highly appreciate the reviewer's instructive comments and suggestions. Next, we reply these comments point-by-point.
>
> - *"Typo: ... the superscript 'ind' should be removed ..."*
>
> Yes, thanks for pointing this out. We updated it in the revision submission.

---

### Official Review · Reviewer_3zzv · 2022-10-30

**Confidence:** 4
**Correctness:** 4
**Technical Novelty And Significance:** 2
**Empirical Novelty And Significance:** 3
**Recommendation:** 8

**Clarity, Quality, Novelty And Reproducibility:**

The paper is well-written, in my opinion, and it is organized effectively for ease of reading. Despite the fact that the technique is not new, the proposed goal is intriguing and may have real-world implications.

**Details Of Ethics Concerns:**

I do not find any concerns.

**Strength And Weaknesses:**

Strength
* The new objective is strongly supported by practical applications.
* Numerical simulations with real-world data are offered to verify the proposed algorithm
* The paper is written well.

Weaknesses
* It seems to me the key technique to lower the communication complexity from $O(\log T)$ to $O(\log \log T)$ is the doubling technique or observation-buffering broadcast mentioned in the paper, which is a well-known technique in bandit literature. It would be preferable if the author(s) could explain in the study why establishing the intended outcomes was challenging.
--------------------

After rebuttal

The author(s) have added discussions regarding the challenge raised during the proof of the communication complexity. And I have raised my score.

**Summary Of The Paper:**

This paper studies distributed multi-armed bandits with the goal of minimizing the largest regret incured among all the agents. This work makes two contributions: First, it suggests a different objective—minimizing individual regret rather than group regret—motivated by real-world applications. Second, it proposes a new algorithm—UCB-TCOM—with almost optimal bound for both individual and group regret. In addition, the new approach only incurs communication cost of $O(\log \log T)$ times.

**Summary Of The Review:**

This paper is well-motivated by real-world applications. The author(s) have proposed a new algorithm with nearly-optimal individual and group regret. Besides, it incurs only $O(\log \log T)$ communication times. However, the techniques used are not brand-new. I struggle to understand how challenging it is to establish the theory in the paper.

---

> ### Author Response · Authors · 2022-11-10
> **On the challenge of establishing our results**
>
> We first want to thank the reviewer's positive feedback.  In this rebuttal submission, we updated a discussion on the analysis challenges of $\texttt{UCBTCOM}$ in Appendix B.1 (the main paper has a pointer at Remark 3 towards this discussion).
> We quote the text as follows (in (2) we illustrate the challenge of applying the doubling phase technique):
>
> >$\texttt{TCOM}$ utilizes two techniques: communication arm set construction ($\S3.1.1$) and observation-buffering broadcast ($\S3.1.2$).
> Each technique alone can reduce the communication times to $O(\log T)$,
> and $\texttt{TCOM}$ combines both to further reduce the communication times to $O(\log (\log T))$.
> This $O(\log (\log T))$ result comes directly from the $\texttt{TCOM}$ policy design.
> The major challenge in the analysis is to establish that---with these reduced and delayed sharing observations due to $\texttt{TCOM}$---the $\texttt{UCBTCOM}$ algorithm still preserves the near-optimal group and individual regrets.
> To do so, we show (1) the communication arm set construction technique can prevent communicating most optimal arm observations while allow communicating most suboptimal arm observations;
> (2) the observation sharing delay due to the observation-buffering broadcast technique (upon the communication arm set technique) does not have an intrinsic impact on $\texttt{UCBTCOM}$'s group and individual regret performance.
>
> >(1) Although the communication arm set construction technique shares the high-level idea of $\texttt{ComEx}$ (Madhushani & Leonard, 2021),
> the analysis of $\texttt{ComEx}$ is not applicable to $\texttt{TCOM}$.
> Because the empirical means utilized for identifying the suboptimal arms in $\texttt{ComEx}$ is only a special case of the tunable confidence intervals that $\texttt{TCOM}$ relies on.
> Therefore, to show that agents can selectively share suboptimal arms' observations by the communication arm set technique, we prove two new results: (a) the optimal arm's observations are often not broadcast (i.e., the optimal arm is often not in the communication arm set, Theorem 2(ii)); (b) suboptimal arm observations are almost always broadcast (i.e.,  the suboptimal arms are often in the set, Theorem 2(iii)).
> The (a) is proved by showing that the number of times that the optimal arm is in the communication arm set is no greater than that of a suboptimal arm (multiplied by a constant factor); thus is upper bounded by $O(\log T)$.
> This is based on revealing a relation between the bandit arm pull policy and the communication arm set construction.
> Then (b) is proved by showing whenever an agent pulls a suboptimal arm, the arm is inside the communication arm set with a high probability, and thus the suboptimal arm's observations are almost always buffered for later communication.
>
> > (2) Our observation-buffering broadcast mechanism shares a high-level idea of doubling phase technique, but its specific design is different from known ones,
>       and, consequently, its analysis addresses novel challenges.
>       The doubling phase technique was utilized in $\texttt{CMA2B}$ literature, e.g., Boursier849
> & Perchet (2019); Wang et al. (2020b); Shi et al. (2021b),
>       where their arm pulls policies within a phase were either uniformly pulling each arm (Boursier & Perchet, 2019; Wang et al., 2020a) or sticking to several arms (Shi et al., 2021b).
>       However, different from these previous "rigid" arm pull policies, $\texttt{UCBTCOM}$ applies the "flexible" $\texttt{UCB}$ arm pull policy.
>       In order to adapt the doubling phase technique to the $\texttt{UCB}$ policy, we propose the observation-buffering broadcast mechanism which separately buffers the observation of each arm and respectively communicates each arm's compound observations whenever this arm's observation times are doubled (or generally, increased by a $\beta$ factor).
>
> >Since this observation-buffering broadcast is different from known doubling phase algorithms, their existing analysis approaches are not applicable to observation-buffering.
>       Hence, proving that this broadcast mechanism
>       does not intrinsically deteriorate the near-optimal group and individual regrets of $\texttt{UCBTCOM}$ is a unique challenge.
>       More specifically, we prove that the delay due to buffering only makes regrets worse by a constant factor.
>       This is via showing that the observation-buffering mechanism separately buffers each arm's observations, and, therefore, the observation delay of each arm's observations (as well as the pulling times of this arm during the delay) can be respectively upper bounded by at most a constant factor multiplying this arm's total previous pulling times (see (9) of Theorem 1's proof as a formal expression).

---

> > ### Comment · Reviewer_3zzv · 2022-11-25
> > **Response after rebuttal**
> >
> > Thanks the author(s) for the detailed explanation and it makes sense to me. I have raised my score.

---

### Author Response · Authors · 2022-11-10
**Summary of Rebuttal Revision**

We would like to thank all reviewers for their instructive comments and insightful questions!

As a complement to our responses, we also uploaded a rebuttal submission. All the changes were marked as blue in the PDF. Apart from some minor updates, we summarize the major ones as follows:

1. We updated a discussion about our algorithm's analysis challenges in Appendix B.1. This corresponds to Reviewer 3zzv's question.
2. We applied $\texttt{TCOM}$ to Thompson sampling and active elimination algorithms and obtained two new algorithms $\texttt{TS-TCOM}$ and $\texttt{AAE-TCOM}$. Both algorithms' pseudo-codes and numerical simulations were provided in Appendix E. We note that the empirical performance of $\texttt{TS-TCOM}$ and $\texttt{AAE-TCOM}$ is similar to the $\texttt{UCB-TCOM}$ algorithm presented in the main paper. This corresponds to a question of Reviewer coEU and further highlights the generality of $\texttt{TCOM}$.
3. We revised the introduction to better motivate our work, especially the individual regret objective. This corresponds to Reviewer VQR1's suggestions.

Thank you,

Paper4474 Authors

---

### Decision · Program_Chairs · 2023-01-20

**Decision:**

Accept: poster

**Justification For Why Not Higher Score:**

I don't think there's a significant breakthrough in terms of techniques but the problem formulation is nice and cute.

**Justification For Why Not Lower Score:**

There's no reason the paper should be rejected. Hence, it should be accepted.

**Metareview: Summary, Strengths And Weaknesses:**

This paper studies distributed multi-armed bandits with the goal of minimizing the largest regret incurred among all the agents. This is a novel and meaningful objective. The authors proposes an algorithm CMA2B which is shown to be near-optimal and furthermore, the amount of communication is O(log log T) where T is the number of decision rounds.

Generally, the reviewers were convinced about the novelty of the setup and contributions and all recommended acceptance. This paper is a welcome addition to the bandit literature. It would be good if the authors can think about the question of the Pareto optimality (tradeoff) between the regret (individual/group) and the communication overhead. I believe such questions may have been addressed in the bandit-federated learning literature.

**Note From Pc:**

if the above contains the word "oral" or "spotlight" please see: "oral" presentation means -> notable-top-5% and "spotlight" means -> notable-top-25%. As stated in our emails, we are disassociating presentation type from AC recommendations

**Summary Of Ac-Reviewer Meeting:**

NIL